# LaX: Boosting Low-Rank Training of Foundation Models via Latent Crossing

**Ruijie Zhang**[*], **Ziyue Liu**[*], **Zhengyang Wang**, **Zheng Zhang**[†]

University of California at Santa Barbara

{ruijiezhang, ziyueliu, zhengyangwang}@ucsb.edu, zhengzhang@ece.ucsb.edu

## Abstract

Training foundation models such as ViTs and LLMs requires tremendous computing cost. Low-rank matrix or tensor factorization offers a parameter-efficient alternative, but often downgrades performance due to the restricted parameter space. In this work, we introduce **Latent Crossing (LaX)** – a simple yet effective plug-and-play module that enhances the capacity of low-rank models by enabling information flow across low-rank subspaces. We extensively validate the benefits of LaX on pre-training tasks with ViT-Base/Large and LLaMA-like models ranging from 60M to 1B parameters. LaX boosts low-rank model performance to match or exceed the full-rank baselines while using 2-3× fewer parameters. When equipped with low-rank adapters (i.e., LoRA [23]) for fine-tuning LLaMA-7/13B, LaX consistently improves performance on arithmetic and common sense reasoning tasks with negligible cost.

## 1 Introduction

Following neural scaling laws [28, 22, 33], the size and training data of foundation models have grown rapidly, exemplified by models such as ViT-22B [7], GPT-3 (175B) [3], LLaMA-3 (405B) [13], and PaLM (504B) [5]. These large-scale foundation models have achieved remarkable success in diverse applications such as language and vision. However, their success comes at immense computing cost, typically on the scale of multi-million GPU hours per pre-training run. As the unsustainable trend continues, training or even deploying such foundation models has become prohibitively expensive for most research institutions and organizations around the world.

To address these challenges, the community has become increasingly interested in low-rank approximation techniques. This is largely motivated by the empirical observation that weight matrices in deep neural networks often exhibit low effective ranks [1, 11, 27, 64, 43]. Classical matrix compression techniques (such as singular value decomposition (SVD) [10]) and tensor decomposition methods (e.g. Canonical Polyadic (CP), Tensor Train (TT) [16, 4, 29] and Tucker decomposition [51]) have been widely applied to reduce the number of trainable parameters by instantiating and updating the lightweight low-rank factors [31, 45, 12, 57, 60, 56, 39, 65, 34]. These approaches exemplify the paradigm of "low-rank training" and have achieved varying degrees of success. In particular, parameter-efficient fine-tuning (PEFT) [23, 63, 41, 17, 62] has drastically reduced the barrier to fine-tuning large language models while producing competitive results. Recent efforts [38, 66, 15, 60] have extended similar concepts to pre-training. Although low-rank methods typically reduce the model size and computing cost, they introduce a critical trade-off: smaller ranks yield lower capacity and often harm performance, whereas larger ranks incur additional cost, undermining the intended efficiency (see Fig. 1 (a)).

---

[*]Equal contribution

[†]Corresponding Author

39th Conference on Neural Information Processing Systems (NeurIPS 2025).

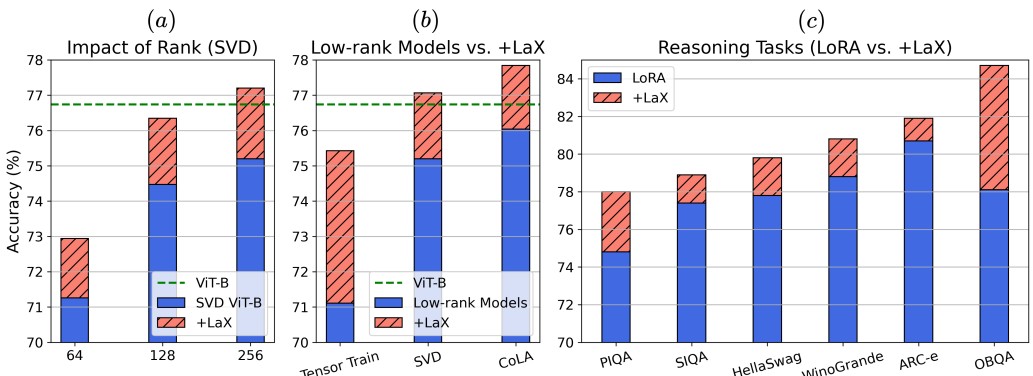

Figure 1: LaX boosts the performance of low-rank training methods. (a) SVD-based pre-training ViT-B on ImageNet-1K with different matrix ranks: lower-rank leads to greater performance drop; LaX consistently improves the performance in all settings. (b) Pre-training ViT-B on ImageNet-1K with different low-rank methods. LaX significantly improves performance for all low-rank methods, even surpassing the full-rank pre-training. (c) Fine-tuning LLaMA-7B on commonsense reasoning tasks using LoRA, with and without LaX respectively. LaX improves LoRA's fine-tuning performance in all tasks.

In this work, we propose **Latent Crossing** (LaX), a lightweight, drop-in module designed to enhance the capacity of low-rank models without explicitly increasing matrix/tensor ranks. By allowing information flow across low-rank subspaces via residual connections, LaX improves model performance while keeping the parameter budget nearly unchanged. Importantly, LaX can be seamlessly integrated with existing low-rank modules such as LoRA [23], SVD, CoLA [43] and TT [45], serving as a **plug-and-play performance booster** that significantly narrows or eventually closes the gap between low-rank and full-rank models.[†]

We summarize our contributions as follows:

1. We propose **LaX**, a lightweight module that increases the capacity of existing low-rank structures without compromising efficiency. By allowing information flow across low-rank subspaces via residual connections, LaX consistently boosts performance in both pretraining and fine-tuning settings (Fig. 1).

2. We design **LaX Gate** to align mismatched bottlenecks for low-rank models. To support diverse architectural and computational constraints, we introduce several variants of the LaX gating mechanism, each balancing expressiveness and efficiency under different deployment settings. We also provide practical guidelines for adapting LaX gating to a variety of tasks.

3. We evaluate LaX in a set of low-rank pre-training and fine-tuning experiments for both language and vision foundation models. In ViT pre-training, LaX improves accuracy by up to $4.32\%$ on ImageNet-1K. For LLM pre-training, LaX shows **consistent gains** across various model scales and different low-rank architectures. When combined with LoRA for fine-tuning, LaX enhances reasoning capabilities of LLaMA-7B/13B on both arithmetic and commonsense reasoning tasks.

## 2 Related Works

### 2.1 Low-rank Factorization for Neural Networks

To mitigate the high computational and storage costs associated with large models, low-rank factorizations have been widely explored as an effective strategy [47, 1]. Early efforts focused on applying low-rank matrix factorization, such as SVD, to compress layers [8, 26, 31, 30, 61]. More recently, LoRA-style adapters [23, 63, 41, 17, 62] extend this idea by adapting SVD-like modules onto frozen pre-trained weights, enabling efficient fine-tuning of large foundation models. Besides, low-rank tensor factorization, including tensor train (TT) [46, 6, 54], Canonical Polyadic (CP) [16, 4, 29], Tucker decomposition [51], and other tensor-based formats [67, 32, 2, 50] have shown promise for reducing complexity of models [40, 36, 52, 42, 39, 65, 34, 60]. In this paper, we focus on representative methods from both directions:

---

[†]We provide our code here

**Low-rank Matrix Factorization.** SVD factorizes a weight matrix $\mathbf{W} \in \mathbb{R}^{d_{\text{out}} \times d_{\text{in}}}$ as $\mathbf{W} = \mathbf{BA}$, where $\mathbf{A} \in \mathbb{R}^{r \times d_{\text{in}}}$ and $\mathbf{B} \in \mathbb{R}^{d_{\text{out}} \times r}$, resulting in a reduced parameter count of $r(d_{\text{in}} + d_{\text{out}})$. CoLA [43] further extends this factorization to an autoencoder by injecting a nonlinear activation $\sigma$ between $\mathbf{A}$ and $\mathbf{B}$, replacing a linear layer $\mathbf{Wx}$ with the bottleneck structure $\mathbf{B}\,\sigma(\mathbf{Ax})$.

**Low-rank Tensor Factorization.** We adopt tensor train decomposition [46] as a representative higher-order factorization method. It reshapes a weight matrix $\mathbf{W} \in \mathbb{R}^{d_{\text{out}} \times d_{\text{in}}}$ into an order-$n$ tensor $\boldsymbol{\mathcal{W}} \in \mathbb{R}^{d_0 \times d_2 \cdots \times d_{n-1}}$ with $d_{\text{out}} \times d_{\text{in}} = \prod_{i=0}^{n-1} d_i$, and decomposes it with a sequence of tensor cores $\{\boldsymbol{\mathcal{C}}^0, \boldsymbol{\mathcal{C}}^1, \ldots, \boldsymbol{\mathcal{C}}^{n-1}\}$, where each $\boldsymbol{\mathcal{C}}^i \in \mathbb{R}^{r_i \times d_i \times r_{i+1}}$. The weight is represented via a chain of tensor contractions:

$$\boldsymbol{\mathcal{W}} = \boldsymbol{\mathcal{C}}^0 \times_{3,1} \boldsymbol{\mathcal{C}}^1 \times_{3,1} \cdots \times_{3,1} \boldsymbol{\mathcal{C}}^{n-1}.$$

However, the extent of parameter reduction, as well as its benefits heavily depend on the chosen rank. Lower-rank settings often suffer from a loss of expressiveness and degrade model performance [66, 43, 23, 59, 47, 1], while higher ranks reintroduce computational overhead. In this work, we propose a simple, plug-and-play module that complements general low-rank training methods in neural networks, aiming to **recover lost performance without compromising their efficiency**.

## 2.2 Residual Mechanism

ResNet [18] stands as one of the most influential milestones in deep learning by introducing a skip connection that routes a layer's input directly to its output. This *residual mechanism* mitigates the vanishing gradients and enables the stable training of deep networks with hundreds of layers. This design principle has been broadly adopted in numerous architectures, including recurrent neural networks [14], transformers [9, 53], and diffusion-based models [21]. In addition to its empirical success, the authors provided a theoretical justification for the residual connection [19]. Building on this foundation, subsequent research has proposed various improvements and theoretical analyses to further enhance the residual learning paradigm [58, 20, 55, 35], consistently emphasizing the central role of residual pathways in improving both convergence and generalization in deep networks.

Inspired by ResNet, we propose **Latent Crossing (LaX)**. LaX serves as a model performance booster by enabling information flow across low-rank subspaces, restoring expressiveness often lost due to rank constraints. Across a wide range of tasks, LaX delivers consistent performance gains while preserving the efficiency advantages of low-rank architectures.

## 3 The LaX Method

Our goal is to augment existing low-rank models with a lightweight module that recovers the performance typically lost due to the low-rank constraints. In Section 3.1, we present the background and motivation behind this work. Following this, Section 3.2 introduces the design of the LaX module for different low-rank structures, Section 3.3 introduces LaX Gate and outlines its key variants, and Section 3.5 offers practical guidelines to facilitate its integration into a wide range of low-rank training frameworks.

## 3.1 Latent Crossing

As discussed in Section 2.1, given a weight matrix $\mathbf{W} \in \mathbb{R}^{d_{\text{out}} \times d_{\text{in}}}$ from an arbitrary linear layer, low-rank methods approximate it as low-rank factors. We take SVD as a motivating example to illustrate this concept. Let $\mathbf{x}_i \in \mathbb{R}^{d_{\text{in}}}$ denote the input to the $i$-th low-rank layer. A down-projection matrix $\mathbf{A}_i \in \mathbb{R}^{r \times d_{\text{in}}}$ maps $\mathbf{x}_i$ into a lower-dimensional latent representation $\mathbf{h}_i \in \mathbb{R}^r$, which is subsequently transformed back to the output space using an up-projection matrix $\mathbf{B}_i \in \mathbb{R}^{d_{\text{out}} \times r}$:

$$\mathbf{h}_i = \mathbf{A}_i \mathbf{x}_i \in \mathbb{R}^r, \quad \mathbf{y}_i = \mathbf{B}\big(\mathbf{A}_i \mathbf{x}_i\big) = \mathbf{B}_i \mathbf{h}_i \in \mathbb{R}^{d_{\text{out}}}. \tag{1}$$

This factorization reduces the number of parameters by choosing a smaller rank $r$ and compresses input into a narrow latent space, which can lead to information bottlenecks, often resulting in a drop of performance due to the constrained searching space. Empirically, increasing the rank $r$ typically improves performance but diminishes the efficiency benefits of the low-rank approach due to increased parameter count and computation.

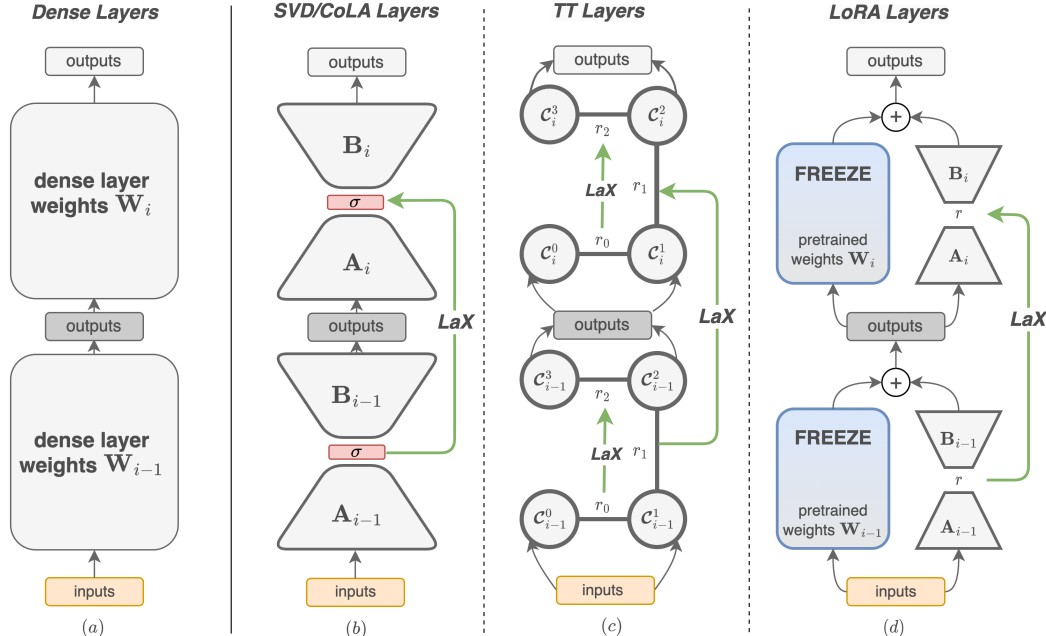

Figure 2: LaX is a general module that can be plugged into low-rank neural network models. **(a)** Dense layers: full information flow, effective but computationally expensive. **(b)** SVD/CoLA[43] layers: rank-$r$ bottlenecks with two factors; LaX can be inserted into the latent space between layers. **(c)** Tensor-train layers: bottleneck structure with four tensor cores, where data flow is governed by tensor contractions; LaX can be applied either between cores or across layers. **(d)** LoRA adapters: LaX can be placed between different adapters.

Our motivation is that, can we **improve the performance of low-rank layers without explicitly increasing the physical rank** $r$**?** Our answer is yes. Instead of directly applying the up-projection $\mathbf{B}_i$ to $\mathbf{h}_i$, LaX incorporates latent features from the previous layer into the up-projection process. Formally, we propose LaX as follows:

$$\mathbf{h}_{i-1} = \mathbf{A}_{i-1}\mathbf{x}_{i-1}, \quad \mathbf{h}_i = \mathbf{A}_i\mathbf{x}_i \in \mathbb{R}^r,$$
$$\tilde{\mathbf{y}}_i = \mathbf{B}_i(\mathbf{h}_i + \mathbf{h}_{i-1}) \in \mathbb{R}^{d_{\text{out}}}. \tag{2}$$

Equivalently, if we stack inputs as $\tilde{\mathbf{x}}_i := \begin{bmatrix} \mathbf{x}_i \\ \mathbf{x}_{i-1} \end{bmatrix} \in \mathbb{R}^{2d_{\text{in}}}$, then LaX can be formulated as

$$\tilde{\mathbf{y}}_i = \mathbf{W}_i^{(LaX)}\tilde{\mathbf{x}}_i, \qquad \boxed{\mathbf{W}_i^{(LaX)} := \begin{bmatrix} \mathbf{B}_i\mathbf{A}_i & \mathbf{B}_i\mathbf{A}_{i-1} \end{bmatrix} \in \mathbb{R}^{d_{\text{out}} \times 2d_{\text{in}}}.} \tag{3}$$

Since $\mathbf{h}_{i-1}$ is naturally produced by the preceding layer during the forward pass, this implicit reuse of intermediate representations facilitates direct information flow across consecutive low-rank projections, requiring no additional parameters or computation overhead.

### 3.2 Variants of LaX

LaX is a general module that is widely applicable to low-rank structures. Eq (2) mainly describes its implementation on matrix factorization methods, where LaX is applied across two consecutive layers. We also refer to this implementation as **Inter-Layer LaX**. In modern architectures such as the transformer, we apply Inter-Layer LaX between the same type of layers across transformer blocks, i.e., from attention (QKV projection) to attention, and from MLP to MLP. This design preserves structural and semantic alignment in residuals while avoiding cross-type interference.

For more fine-grained low-rank structure, such as tensor factorization methods, LaX is not limited to cross-layer only. Take the tensor-train representation in Fig. 3 as an example: $\mathbf{W}$ (dropping layer index for simplicity) is approximated by 6 low-rank factors $\{\mathcal{C}^0, \mathcal{C}^1, \ldots, \mathcal{C}^5\}$, therefore, a series of latent features will be produced when sequentially contracting each factor, such as $\mathcal{C}^0\mathbf{x}$, $\mathcal{C}^0\mathcal{C}^1\mathbf{x}$, $\mathcal{C}^0\mathcal{C}^1\mathcal{C}^2\mathbf{x}$, etc. Along this contraction sequence, earlier results can be used as residuals to form multiple LaX pathways. We refer to this implementation as **Intra-Layer LaX**.

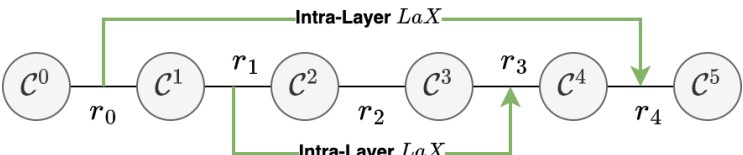

Figure 3: A 6-core Tensor Train layer with the symmetric setting. For Tensor Train layers with identical input and output shapes, we can naturally arrange the tensor ranks in a symmetric configuration, where $r_0 = r_4$ and $r_1 = r_3$ in this example. This reduces the need for shape transformation operations, making **Intra-Layer** LaX more efficient when applied.

### 3.3 LaX Gates

When the latent dimensions are aligned (e.g., SVD with the same rank between layers, symmetric TT ranks within a layer), direct residual pathways can be formed without introducing extra parameters. For example, in QKV projection layers with symmetric TT setup (Fig.3), we configure to ensure residual compatibility. However, a direct addition becomes infeasible when latent features have mismatched dimensions. To handle this, we introduce LaX Gate, a module that aligns and modulates latent features before residual fusion (Fig. 4). The gated residual formulation becomes:

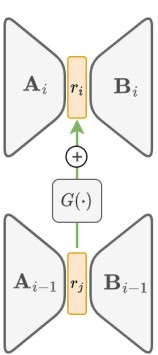

Figure 4: LaX Gate.

$$\tilde{\mathbf{y}}_i = \mathbf{W}_i^{(LaX)} \tilde{\mathbf{x}}_i, \qquad \mathbf{W}_i^{(LaX)} := \begin{bmatrix} \mathbf{B}_i \mathbf{A}_i & \mathbf{B}_i G_i \mathbf{A}_{i-1} \end{bmatrix}. \qquad (4)$$

To accommodate different architectural needs and promote the versatility of LaX, we unify the notion by introducing the following Gate variants:

- **Identity Gate**: Passes latent features through a direct addition without introducing additional parameters, i.e. $G(\cdot) = 1$.

- **Linear Gate**: Introduces a single trainable parameter to control how much information is passed forward, i.e. $G(\cdot) = \beta$, where $\beta \in \mathbb{R}$ is a trainable parameter.

- **Tensor Gate**: As illustrated in Fig. 5, this variant first folds the latent feature vector into a tensor $\mathcal{R} \in \mathbb{R}^{r_0 \times 1 \times r_1}$, then contracts it with two learnable gate tensor cores: $\mathcal{C}^0 \in \mathbb{R}^{1 \times r_0' \times r_0}$ and $\mathcal{C}^1 \in \mathbb{R}^{r_1 \times r_1' \times 1}$, i.e. $\mathcal{R}' = \mathcal{C}^0 \times_{3,1} \mathcal{R} \times_{3,1} \mathcal{C}^1 \times_{3,1} \in \mathbb{R}^{r_0' \times 1 \times r_1'}$ to match targeting shape $r_0' \times r_1'$. In the two-core setting, each gating core includes a singleton dimension. This dimension can be generalized to larger sizes when extending the design to more than two gating cores, allowing for greater flexibility across different use cases.

- **Dense Gate**: Passes latent feature using a $G \in \mathbb{R}^{r \times r}$ linear layer.

We remark that the overhead introduced by LaX in terms of parameter count and computation is often minimal, since the latent rank r is relatively small (e.g., 64 or 128) in low-rank models. In addition to addressing dimensional misalignment, we empirically find that LaX Gate can further boost performance with negligible parameter overhead (see details in Section 4.1 for ViT pre-training with Tensor Gate). Therefore we also experiment LaX Gate on scenarios where dimensions are matched.

### 3.4 Feature Normalization

Additionally, following the common practice of normalizing features after a residual connection, we postpend a *Layer Normalization* (LN) to each LaX pathway. With this normalization, the final form of LaX is as follows:

$$\tilde{\mathbf{y}}_i = \mathrm{LN}(\mathbf{W}_i^{(LaX)} \tilde{\mathbf{x}}_i), \qquad \mathbf{W}_i^{(LaX)} := \begin{bmatrix} \mathbf{B}_i \mathbf{A}_i & \mathbf{B}_i G_{i-1} \mathbf{A}_{i-1} \end{bmatrix}. \qquad (5)$$

### 3.5 Practical Guideline

Here, we provide practical guidelines for selecting the appropriate LaX variant based on empirical observations across different tasks:

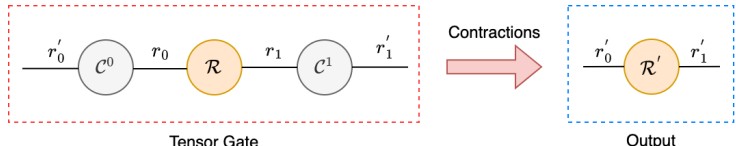

Figure 5: Two-Core Tensor Gate. A residual tensor $\mathcal{R} \in \mathbb{R}^{r_0 \times 1 \times r_1}$ is contracted with two gating tensor cores, $\mathcal{C}^0$ and $\mathcal{C}^1$, producing a transformed residual tensor $\mathcal{R}' \in \mathbb{R}^{r'_0 \times 1 \times r'_1}$.

| Method | Variant | ViT-B | | ViT-L | |
|---|---|---|---|---|---|
| | | # Params ($M$) | Accuracy (%) | # Params ($M$) | Accuracy (%) |
| Original | - | 86.56 | 76.74 | 304.33 | 77.10 |
| SVD | Base Model | 44.17 | 75.20 | 115.77 | 76.81 |
| | + LaX (Ours) | 44.24 | **77.20** (+2.00) | 115.92 | **78.60** (+1.79) |
| Tensor Train | Base Model | 41.18 | 71.11 | 101.97 | 75.21 |
| | + LaX (Ours) | 41.44 | **75.43** (+4.32) | 102.10 | **77.77** (+2.56) |
| CoLA | Base Model | 44.17 | 76.04 | 115.77 | 77.63 |
| | + LaX (Ours) | 44.24 | **77.84** (+1.80) | 115.92 | **79.07** (+1.44) |

Table 1: Accuracy comparison of pre-training on ImageNet-1k datasets. LaX consistently improves pre-training performance across various low-rank models and scales. When applied to CoLA [43], CoLA+LaX achieves the highest accuracy on both ViT-B and ViT-L. Tensor Train models observe the largest gains, with improvements of +4.32%/+2.56% on ViT-B/L.

- **ViTs Pre-training Task**: Vision Transformer pre-training typically involves multiple epochs over medium-sized datasets. Under this setting, we observe (see Tab. 2) that the **Tensor Gate** consistently outperforms other gate variants. We therefore recommend using the **Tensor Gate** for ViT pre-training tasks.

- **LLMs Pre-training Task**: In contrast to ViTs, large language model pre-training typically involves processing a significantly larger number of training tokens across a vast semantic space (e.g., a vocabulary size of 32,000 in LLaMA-1/2), often without completing a full training epoch. In such case, we empirically find that **Identity Gate** off-the-shelf provides consistent and significant improvements to different low-rank architectures, with zero parameter overhead and only negligible compute overhead (see Section 4.2).

- **Fine-Tuning Task**: The optimization space of fine-tuning is already constrained and therefore very small, where introducing additional parameters is often unnecessary. In these scenarios, we recommend using the **Identity Gate** or **Linear Gate** to preserve training efficiency while enabling information flow across low-rank subspaces.

## 4 Pre-training Experiments

We first evaluate the performance of LaX in some low-rank pre-training experiments of ViTs/LLMs.

### 4.1 Pre-training Vision Transformers

We pretrain ViT-Base/Large ($224$ resolution with $16 \times 16$ patch size) and its corresponding low-rank variants on ImageNet-1K [49]. We consider SVD, Tensor Train and CoLA [43] as the baseline low-rank models. All models are trained from scratch for 300 epochs according to the setting of [9]. For SVD and CoLA, we only place Inter-layer LaX with the **Tensor Gate**. For tensor train, we use both Inter-Layer LaX and Intra-Layer LaX with the **Tensor Gate**. More details of the model and training configurations are provided in Appendix A.1.

As shown in Tab. 1, all low-rank baselines experienced accuracy drop compared to the full-rank training. However, LaX consistently improves performance across the three baseline methods (see the training curve in Fig. 6; additional curves in Appendix A.2), with negligible parameter

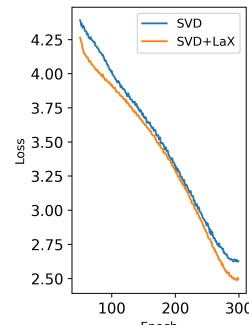

Figure 6: Training Loss

| | rank=256 | | rank=128 | | rank=64 | |
|---|---|---|---|---|---|---|
| **Gate** | **# Params (M)** | **Accuracy (%)** | **# Params (M)** | **Accuracy (%)** | **# Params (M)** | **Accuracy (%)** |
| Base Model | 44.17 | 75.20 | 22.94 | 74.47 | 12.32 | 71.26 |
| **+ Identity Gate** | 44.17 | 75.81 (+0.61) | 22.94 | 74.65 (+0.18) | 12.32 | 71.64 (+0.38) |
| **+ Linear Gate** | 44.17 | 76.11 (+0.91) | 22.94 | 75.31 (+0.84) | 12.32 | 71.72 (+0.46) |
| **+ Tensor Gate** | 44.24 | **77.20** (+2.00) | 22.97 | **76.35** (+1.88) | 12.34 | **72.94** (+1.68) |
| **+ Dense Gate** | 48.55 | 77.03 (+1.83) | 24.04 | 75.43 (+0.96) | 12.60 | 72.33 (+1.07) |

Table 2: Pre-training performance of different LaX Gates on SVD-based ViT-B under varying rank settings. LaX consistently improves performance across all configurations, with the **Tensor Gate** achieving the largest gains while incurring minimal parameter overhead.

overhead ($\leq 0.2\%$). On the ViT-B scale, it recovers the lost performance, boosting accuracy by **+2.00%** to 77.20% for SVD. For Tensor Train, LaX lifts accuracy from 71.11% to 75.43% (**+4.32%**), turning the weakest model into a strong contender. Even for CoLA, the best-performing baseline, LaX adds boosts the accuracy by **1.80%**, reaching 77.84%. Similar trends are observed in ViT-L.

We further evaluate the impact of different LaX Gate variants under varying rank $r$ configurations in ViT pre-training. As shown in Tab. 2, all gate variants consistently improve accuracy across different rank settings compared to their respective base models. Among them, **Tensor Gate** achieves the highest gains with minimal parameter overhead. At rank $256$, **Tensor Gate** improves accuracy by +2.00% with only +0.07M additional parameters. As the rank decreases, the added parameter cost also diminishes: at rank $128/64$, the **Tensor Gate** requires only +0.03M/+0.02M more parameters to achieve +1.88%/+1.68% accuracy gains, respectively. Additional experiments can be found in Appendix B.

| **Model** | **Computation Complexity** |
|---|---|
| Original | $\mathcal{O}(nd^2 + n^2d)$ |
| SVD / CoLA | $\mathcal{O}(ndr + n^2d)$ |
| Tensor Train | $\mathcal{O}(ndr + n^2d)^{\dagger}$ |

Table 3: Model Complexity (per block) under batch size 1.

| **Gate Type** | **FLOPs Overhead** |
|---|---|
| Res / Norm | $\mathcal{O}(nr)$ |
| Identity | $\mathcal{O}(1)$ |
| Linear | $\mathcal{O}(nr)$ |
| Tensor | $\mathcal{O}(nr)$ |
| Dense | $\mathcal{O}(nr^2)$ |

Table 4: LaX Gate Overhead under batch size 1.

**Complexity Analysis**  We further analyzed the computational complexities (measured by the number of FLOPs in each transformer block) of the original models and the gating mechanisms of LaX. Tab. 3 shows the FLOPs of the models without LaX while Tab. 4 shows the additional FLOPs required by LaX gating, where $n$ is the sequence length, $d$ is the hidden dimension, and $r$ is the rank. As shown by the tables, the computation overhead introduced by *Res*, *Norm*, *Identity*, *Linear*, and *Tensor Gate* is at least an order of magnitude smaller than the original models, and so negligible. *Dense Gate* introduces overhead that is quadratic in $r$, but it could still be acceptable if $r << d$.

## 4.2   Pre-training Language Models

We further evaluate LaX in language model pre-training tasks where previous work suggests that pure low-rank architectures often cause performance drop [37, 66, 15]. More recent work such as CoLA [43] and LORO [44], have shown promising results by imposing low-rank activations or performing manifold optimization. Since LORO optimizes $\mathbf{A}$ and $\mathbf{B}$ in the rank-$r$ manifold that $\mathbf{W} = \mathbf{BA}$ lies on, the proposed formulation of LaX contradicts this assumption. Consequently, we compare LaX with SVD and CoLA$^{\dagger}$, and directly cite the results reported in [66, 15, 44, 43].

**We adopt the same experimental setup** from recent benchmarks [66, 15, 44, 43], pre-training LLaMA-like models from 60M to 1B parameters on C4 [48] without data repetition and using compute-optimal token-to-parameter ratios$^{\dagger}$. All linear layers in the original LLaMA architecture are replaced with low-rank layers. For CoLA, we follow [43], and implement the SVD baseline by removing its low-rank activations and/or restoring the original activation. All methods use the same rank for fairness. Full training details are provided in Appendix A.3.

---

$^{\dagger}$This provides a loose upper bound on complexity, but still tighter than the one reported in [45].

$^{\dagger}$Due to resource constraint, we focus on baseline architectures that performed better in our ViT experiments

$^{\dagger}$The token-to-parameter (T2P) ratios are roughly compute optimal [22].

| | 60M | | | 130M | | | 350M | | | 1B | | |
|---|---|---|---|---|---|---|---|---|---|---|---|---|
| *r / d*
*Tokens* | 128 / 512
1.1B | | | 256 / 768
2.2B | | | 256 / 1024
6.4B | | | 512 / 2048
13.1B | | |
| | PPL | Param | Mem | PPL | Param | Mem | PPL | Param | Mem | PPL | Param | Mem |
| Full-rank | 34.06 | 58 | 0.43 | 24.36 | 134 | 1.00 | 18.80 | 368 | 2.74 | 15.56 | 1339 | 9.98 |
| ReLoRA [37] | 37.04 | 58 | 0.37 | 29.37 | 134 | 0.86 | 29.08 | 368 | 1.94 | 18.33 | 1339 | 6.79 |
| GaLore [66] | 34.88 | 58 | 0.36 | 25.36 | 134 | 0.79 | 18.95 | 368 | 1.90 | 15.64 | 1339 | 6.60 |
| SLTrain [15] | 34.15 | 44 | 0.32 | 26.04 | 97 | 0.72 | 19.42 | 194 | 1.45 | 16.14 | 646 | 4.81 |
| LORO [44] | 33.96 | 43 | 0.32 | 24.59 | 94 | 0.70 | 18.84 | 185 | 1.38 | 15.19 | 609 | 4.54 |
| SVD | 36.25 | 43 | 0.32 | 26.84 | 94 | 0.70 | 21.18 | 185 | 1.38 | 16.54 | 609 | 4.54 |
| SVD + LaX | **33.54**
(-2.71) | 44 | 0.33 | 24.63
(-2.21) | 94 | 0.70 | 18.90
(-2.28) | 185 | 1.38 | 15.51
(-1.03) | 609 | 4.54 |
| CoLA [43] | 34.04 | 43 | 0.32 | 24.48 | 94 | 0.70 | 19.40 | 185 | 1.38 | 15.52 | 609 | 4.54 |
| CoLA + LaX | **33.21**
(-0.83) | 44 | 0.33 | **24.21**
(-0.27) | 99 | 0.74 | **18.51**
(-0.89) | 196 | 1.46 | **14.78**
(-0.74) | 609 | 4.54 |

Table 5: Comparisons of LaX and its base models against other low-rank methods on pre-training C4 dataset [48] from 60M to 1B. We report the validation perplexity (PPL (↓)), number of parameters in millions (Param), and the estimated total memory usage in GB (Mem) excluding activations based on BF16 precision. Results other than LaX and vanilla SVD are from [66, 15, 43, 44].

| | *LaX* | | 60M | | 130M | | 350M | |
|---|---|---|---|---|---|---|---|---|
| | Res | Gate | PPL | Rank | PPL | Rank | PPL | Rank |
| Full-Rank | - | - | 34.06 | 512 | 24.36 | 768 | 18.80 | 1024 |
| LORO | - | - | 33.96 | 128 | 24.59 | 256 | 18.84 | 256 |
| SVD –
Lower Rank | ✗
✓
✓ | ✗
✗
✓ | 36.25
33.61 (-2.64)
33.54 (-2.71) | 128 | 26.84
24.63 (-2.21)
24.66 (-2.18) | 256 | 21.18
18.90 (-2.28)
18.93 (-2.25) | 256 |
| CoLA –
Lower Rank | ✗
✓
✓ | ✗
✗
✓ | 34.04
33.82 (-0.22)
**33.21** (-0.83) | 128 | 24.48
24.37 (-0.11)
**24.21** (-0.27) | 256 | 19.40
18.81 (-0.59)
**18.51** (-0.89) | 256 |
| SVD –
Higher Rank | ✗
✓
✓ | ✗
✗
✓ | 33.45
31.62 (-1.83)
31.82 (-1.63) | 224 | 26.20
23.86 (-2.34)
23.97 (-2.23) | $\begin{bmatrix}256\\384\end{bmatrix}$ | 19.68
18.39 (-1.29)
18.21 (-1.47) | $\begin{bmatrix}384\\512\end{bmatrix}$ |
| CoLA –
Higher Rank | ✗
✓
✓ | ✗
✗
✓ | 31.52
31.42 (-0.10)
**30.90** (-0.60) | 224 | 23.97
23.74 (-0.23)
**23.42** (-0.55) | $\begin{bmatrix}256\\384\end{bmatrix}$ | 18.32
17.53 (-0.79)
**17.34** (-0.98) | $\begin{bmatrix}384\\512\end{bmatrix}$ |

Table 6: Comparisons of LaX on SVD/CoLA between different Gate variants (✗ denotes Identity Gate, ✓ denotes Dense Gate) and rank choices across 60M to 350M scales. For scenarios where a vector of ranks is provided, smaller one is for attention layers and the larger one is for MLP layers.

As shown in Tab. 5, LaX improves the validation perplexity of SVD and CoLA across all scales. In particular, vanilla SVD performs poorly compared to most baselines but can be boosted to perform on par with or surpassing LORO and CoLA. While CoLA perform similarly to LORO, its LaX -boosted version surpasses LORO on all scales. In addition, LaX just uses the standard Adam optimizer and does not need LORO's complex manifold gradient computations and deeply customized training strategies[†].

Tab. 6 compares SVD/CoLA variants across different ranks and LaX configurations. From Tab. 6 we observe that with either Identity Gate (denoted by ✗) or Dense Gate (denoted by ✓), LaX continues to increase performance. In particular, the results in Tab. 6 further demonstrate that LaX is consistently effective regardless of whether the base model has a higher rank. The trend continues to hold that LaX boosts more on a weaker base model than on a stronger base model.

The only mixed message in Tab. 6 is the effectiveness of LaX Gate. In CoLA from 60M to 350M, the Dense Gate consistently outperforms the Identity Gate. However, for the SVD method, a Dense Gate does not provide a consistent benefit. When results are mixed, the rule-of-thumb is to be conservative; therefore, we recommend using **Identity Gate** in language model pre-training, as it

---

[†]LORO requires periodical computations of manifold gradient which involves tuning the update frequency, at each update step a learning rate warm-up and a refreshment of Adam statistics.

| Model | Method ($r = 32$) | # Params (%) | MultiArith | GSM8K | AddSub | AQuA | SingleEq | SVAMP | Avg |
|---|---|---|---|---|---|---|---|---|---|
| LLaMA-7B | LoRA | 0.83 | 95.0 | 36.1 | 84.3 | 17.7 | 84.4 | 51.8 | 61.6 |
| | LoRA + LaX (Ours) | 0.83 | **96.9** (+1.9) | **37.7** (+1.6) | **84.8** (+0.5) | **19.3** (+1.6) | **87.8** (+3.4) | **53.6** (+1.8) | **63.4** (+1.8) |
| LLaMA-13B | LoRA | 0.67 | 95.2 | 47.5 | 86.0 | 18.2 | 89.8 | 54.6 | 65.2 |
| | LoRA + LaX (Ours) | 0.67 | **97.3** (+2.1) | **49.0** (+1.5) | **86.3** (+0.3) | **20.9** (+2.7) | **91.9** (+2.1) | **58.3** (+3.7) | **67.3** (+2.1) |

Table 7: Accuracy comparison of LoRA and LaX-LoRA on six math reasoning datasets.

| Model | Method ($r = 32$) | # Params (%) | BoolQ | PIQA | SIQA | HellaSwag | WinoGrande | ARC-e | ARC-c | OBQA | Avg |
|---|---|---|---|---|---|---|---|---|---|---|---|
| LLaMA-7B | LoRA | 0.83 | 68.9 | 80.7 | 77.4 | 78.1 | 78.8 | 77.8 | 61.3 | 74.8 | 74.7 |
| | LoRA + LaX (Ours) | 0.83 | **69.6** (+0.7) | **81.9** (+1.2) | **78.9** (+1.5) | **84.7** (+6.6) | **80.8** (+2.0) | **79.8** (+2.0) | **64.8** (+3.5) | **78.0** (+3.2) | **77.3** (+2.6) |
| LLaMA-13B | LoRA | 0.67 | **72.1** | 83.5 | 80.5 | 90.5 | 83.7 | 82.8 | 68.3 | 82.4 | 80.5 |
| | LoRA + LaX (Ours) | 0.67 | 71.3 (-0.8) | **85.4** (+1.9) | **81.3** (+0.8) | **91.3** (+0.8) | **84.1** (+0.4) | **84.4** (+1.6) | **71.8** (+3.5) | **83.1** (+0.7) | **81.6** (+1.1) |

Table 8: Comparison of LoRA and LaX-LoRA on Commonsense Reasoning Benchmarks.

already effectively boosts SVD and CoLA architectures. Consequently, we conduct the pre-training experiments on the 1B scale using the Identity Gate.

On larger-scale CoLA (e.g., 350M and 1B), LaX tends to bring more benefit, contrary to the stable or decreasing trend observed in SVD. This may be caused by the architectural difference between SVD and CoLA, indicating that CoLA might benefit more from LaX when it scales up.

# 5 Fine-tuning Experiments

Finally we show the effectiveness of LaX in low-rank fine-tuning. We incorporate LaX into LoRA (Fig. 2 (d)) and consider two widely used reasoning benchmarks [41, 25]: Arithmetic/Commonsense Reasoning. The fine-tuning configuration in this section strictly follows [24]. Additional configuration details are provided in Appendix A.4.

## 5.1 Arithmetic Reasoning

We fine-tune LLaMA-7B /13B on the Math10K dataset and assess performance in six arithmetic reasoning subtasks. For this evaluation, we configure LaX with Linear Gate.

Tab. 7 shows that augmenting LoRA with LaX yields consistent accuracy improvements across all six arithmetic subtasks for both LLaMA-7B and LLaMA-13B. For the 7B model, the average score improves from 61.6% to 63.4% (+1.8%), while for the 13B model it increases from 65.2% to 67.3% (+2.1%), suggesting that LaX's rank expansion mechanism effectively provides additional representation capacity required by fine-tuning. Even on subtasks where LoRA already performs strongly (e.g. MultiArith, AddSub), LaX delivers consistent improvements.

## 5.2 Commonsense Reasoning

Following [25, 41], we merge the training datasets from all eight commonsense reasoning tasks into a unified training set and evaluate the performance separately on each task. In this experiment, we configure LaX with the Identity Gate. As shown in Tab. 8, LaX consistently outperforms the LoRA baseline in all reasoning tasks. For LLaMA-7B, the average accuracy increases from 74.7% to 77.3% (+2.6%), with a notable gain of +6.6% on HellaSwag. For LLaMA-13B, the overall score rises by +1.1%; only BoolQ exhibits a marginal decline (-0.8%).

# 6 Conclusion

In this work, we have presented **Latent Crossing** (LaX), a lightweight and versatile module designed to improve the training performance of low-rank compressed models. Although low-rank methods are effective in reducing computational overhead, they often suffer from a loss in model expressiveness and performance. LaX addresses this limitation by enabling information flow between low-rank subspaces through residual connections equipped with simple gating mechanisms. As a result, LaX serves as a general plug-in booster that enhances a wide range of low-rank models, across both pretraining and fine-tuning scenarios, and for both language and vision tasks.

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

# A  Hyperparameter

## A.1  ViTs Pre-training Configurations

| Models | LaX | Rank | Epochs | Base LR | LR decay | Weight decay | Dropout | Warmup |
|---|---|---|---|---|---|---|---|---|
| ViT-B | – | – | 300 | 3e-3 | cosine | 0.3 | 0.1 | 10 |
| SVD | × ✓ | 256 | | | | | | |
| TT | × ✓ | 336 | 300 | 1e-3 | cosine | 0.3 | 0.1 | 10 |
| CoLA | × ✓ | 256 | | | | | | |

Table 9: Training hyperparameter for ViT-B and its low-rank variants on ImageNet-1k.

| Models | LaX | Rank | Epochs | Base LR | LR decay | Weight decay | Dropout | Warmup |
|---|---|---|---|---|---|---|---|---|
| ViT-L | – | – | 300 | 3e-3 | cosine | 0.3 | 0.1 | 10 |
| SVD | × ✓ | 256 | | | | | | |
| TT | × ✓ | 256 | 300 | 1e-3 | cosine | 0.3 | 0.1 | 10 |
| CoLA | × ✓ | 256 | | | | | | |

Table 10: Training hyperparameter for ViT-L and its low-rank variants on ImageNet-1k.

| Models | Inter-Layer LaX | Intra-Layer LaX |
|---|---|---|
| SVD | | × |
| TT | Tensor Gate | Tensor Gate |
| CoLA | | × |

Table 11: LaX gating variants for different low-rank methods.

| Models | Inter-Layer LaX | Intra-Layer LaX |
|---|---|---|
| SVD | | $\times$ |
| TT | QKV+MLP | QKV+MLP |
| CoLA | | $\times$ |

Table 12: LaX placed layers for different low-rank methods.

| Models | # Cores | QKV $d_i$ | MLP1 $d_i$ | MLP2 $d_i$ |
|---|---|---|---|---|
| ViT-B | 4 | {32,24,24,32} | {32,24,48,64} | {48,64,24,32} |
| ViT-L | 4 | {32,32,32,32} | {32,32,64,64} | {64,64,32,32} |

Table 13: Tensor Train Dimension Configuration for ViTs.

## A.2 ViT Pre-training Loss Curves

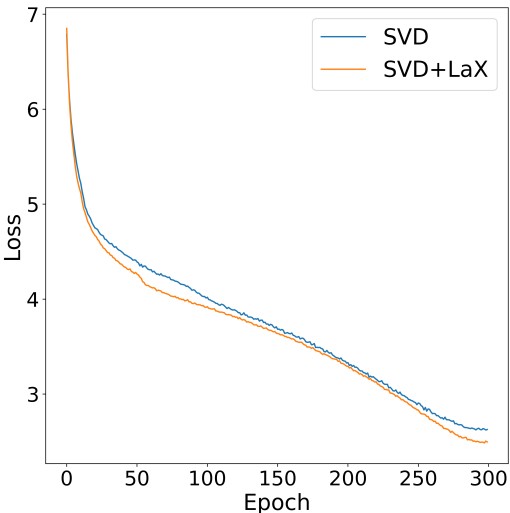

Figure 7: Pre-training Loss Curve of SVD and SVD+LaX on ImageNet-1k

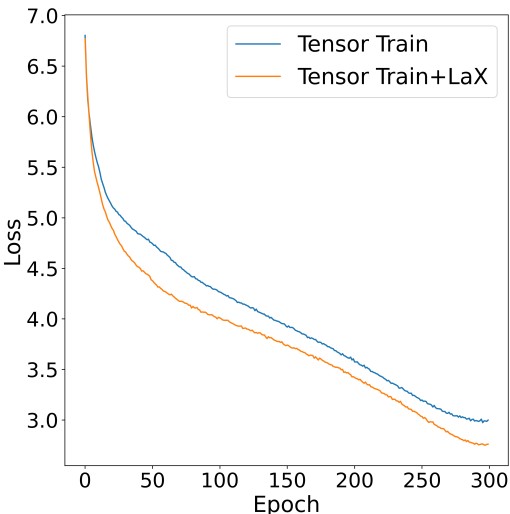

Figure 8: Pre-training Loss Curve of Tensor Train and Tensor Train+LaX on ImageNet-1k

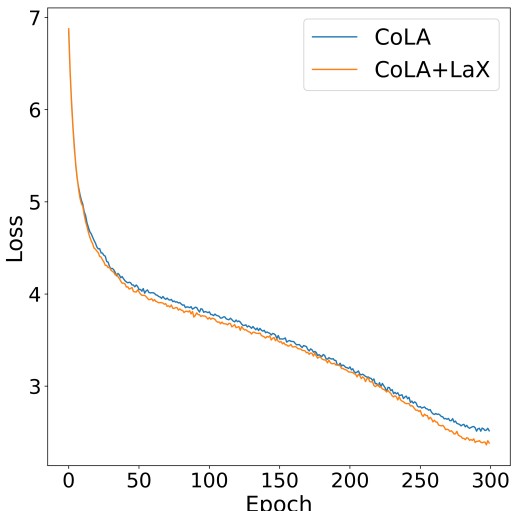

Figure 9: Pre-training Loss Curve of CoLA and CoLA+LaX on ImageNet-1k

### A.3 LLMs Pre-training Configurations

| Scales | Models | LaX | Steps | Base LR | LR decay | Weight decay | Warmup | Gradient clipping |
|---|---|---|---|---|---|---|---|---|
| 60M | SVD | ✗
✓ | 10k | 2e-3
2e-2 | cosine | 0.01 | 2k | 0.5 |
|  | CoLA | ✗
✓ |  | 6e-3
4e-2 |  |  |  |  |
| 130M | SVD | ✗
✓ | 20k | 1e-3
1e-2 | cosine | 0.01 | 4k | 0.5 |
|  | CoLA | ✗
✓ |  | 4e-3
2e-2 |  |  |  |  |
| 350M | SVD | ✗
✓ | 60k | 1e-3
1e-2 | cosine | 0.01 | 12k | 0.5 |
|  | CoLA | ✗
✓ |  | 3e-3
2e-2 |  |  |  |  |
| 1B | SVD | ✗
✓ | 100k | 8e-4
3e-3 | cosine | 0.01 | 20k | 0.5 |
|  | CoLA | ✗
✓ |  | 2e-3
1e-2 |  |  |  |  |

Table 14: Hyper-parameters for pre-training SVD and CoLA and their LaX variants for LLaMA-like models from 60M to 1B.

The primal hyper-parameter for LLM pre-training experiments is the learning rate. For small models such as 60M and 130M, we typically sweep at the scale of 1e-3 for base models and 1e-2 for their LaX variants. The rule-of-thumb that we empirically found is to choose the largest learning rate that does not cause divergence issues. In particular, SVD base models are severely more sensitive to learning rate, and can only afford smaller settings compared to base CoLA. For both SVD and CoLA, LaX offers additional stability that facilitates an order of magnitude larger learning rates. The experience of training smaller models are then adopted for larger scales such as 350M and 1B, which continue following the trend that the proper choice of learning rate decreases when model scale increases. For the same scale, we did not find evident that further tuning learning rates are beneficial. Consequently, we adopt the same setting when only changing the rank for each scale.

## A.4 LaX-LoRA Fine-tuning Configurations

| Hyperparameters (LoRA) | LLaMA-7B | LLaMA-13B |
|---|---|---|
| Rank $r$ | 32 | 32 |
| $\alpha$ | 64 | 64 |
| Dropout | 0.0 | 0.0 |
| Optimizer | AdamW | AdamW |
| LR | 3e-4 | 3e-4 |
| Scheduler | Linear | Linear |
| Batch size | 16 | 16 |
| Accumulation steps | 4 | 4 |
| Cut off length | 256 | 256 |
| Warmup steps | 100 | 100 |
| Epochs | 3 | 3 |
| Where | Q,K,V,Up,Down | Q,K,V,Up,Down |

Table 15: Commonsense Hyperparameter settings for LoRA on LLaMA-7B and LLaMA-13B.

| Hyperparameters (LaX-LoRA) | LLaMA-7B | LLaMA-13B |
|---|---|---|
| Rank $r$ | 32 | 32 |
| $\alpha$ | 64 | 64 |
| Dropout | 0.0 | 0.0 |
| Optimizer | AdamW | AdamW |
| LR | 3e-4 | 3e-4 |
| Scheduler | Linear | Linear |
| Batch size | 16 | 16 |
| Accumulation steps | 4 | 4 |
| Cut off length | 256 | 256 |
| Warmup steps | 100 | 100 |
| Epochs | 3 | 3 |
| Where | Q,K,V,Up,Down | Q,K,V,Up,Down |
| Where LaX | Q,K,V,Up,Down | Q,K,V,Up,Down |
| LaX Gate | Identity | Identity |

Table 16: Commonsense Hyperparameter settings for LaX-LoRA on LLaMA-7B and LLaMA-13B.

| Hyperparameters (LoRA) | LLaMA-7B | LLaMA-13B |
|---|---|---|
| Rank $r$ | 32 | 32 |
| $\alpha$ | 64 | 64 |
| Dropout | 0.0 | 0.0 |
| Optimizer | AdamW | AdamW |
| LR | 3e-4 | 3e-4 |
| Scheduler | Linear | Linear |
| Batch size | 16 | 16 |
| Accumulation steps | 4 | 4 |
| Cut off length | 256 | 256 |
| Warmup steps | 100 | 100 |
| Epochs | 3 | 3 |
| Where | Q,K,V,Up,Down | Q,K,V,Up,Down |

Table 17: Arithmetic Hyperparameter settings for LoRA on LLaMA-7B and LLaMA-13B.

| Hyperparameters (LaX-LoRA) | LLaMA-7B | LLaMA-13B |
|---|---|---|
| Rank $r$ | 32 | 32 |
| $\alpha$ | 64 | 64 |
| Dropout | 0.0 | 0.0 |
| Optimizer | AdamW | AdamW |
| LR | 3e-4 | 3e-4 |
| Scheduler | Linear | Linear |
| Batch size | 16 | 16 |
| Accumulation steps | 4 | 4 |
| Cut off length | 256 | 256 |
| Warmup steps | 100 | 100 |
| Epochs | 3 | 3 |
| Where | Q,K,V,Up,Down | Q,K,V,Up,Down |
| Where LaX | Q,K,V,Up,Down | Q,K,V,Up,Down |
| LaX Gate | Linear | Linear |

Table 18: Arithmetic Hyperparameter settings for LaX-LoRA on LLaMA-7B and LLaMA-13B.

# B  Additional Experiments

## B.1  Where LaX Contributes Mostly

To identify where LaX is most effective, we analyzed the checkpoint from Tab. 2 (rank = 256), focusing on the scalar values of the Linear Gates. Since larger gate values indicate a stronger reliance on LaX, this allows us to quantify the relative contribution of LaX across components.

| LaX Layer | Q | K | V | Proj | FC1 | FC2 |
|---|---|---|---|---|---|---|
| 1 | 3.222 | 3.825 | 0.254 | 0.472 | 2.966 | 0.003 |
| 2 | 1.370 | 1.653 | 0.406 | 0.098 | 2.005 | 0.004 |
| 3 | 1.596 | 1.960 | 0.816 | 0.037 | 1.462 | 0.040 |
| 4 | 1.550 | 1.522 | 1.412 | 0.033 | 1.360 | 0.552 |
| 5 | 2.071 | 2.161 | 0.847 | 0.018 | 1.272 | 0.026 |
| 6 | 2.184 | 2.023 | 1.746 | 0.004 | 1.929 | 0.002 |
| 7 | 0.533 | 1.210 | 2.116 | 0.002 | 2.038 | 0.004 |
| 8 | 0.370 | 1.336 | 1.895 | 0.001 | 1.860 | 0.005 |
| 9 | 0.259 | 1.040 | 1.911 | 0.002 | 1.800 | 0.004 |
| 10 | 0.857 | 0.955 | 1.356 | 0.869 | 2.332 | 0.002 |
| 11 | 1.203 | 2.330 | 2.983 | 1.024 | 2.012 | 1.477 |

Table 19: Scalar values of Linear Gates for each module within LaX layers. Higher values indicate stronger reliance on LaX.

Our analysis reveals several distinct patterns in how LaX contributes across Transformer layers. For the Q and K projections, LaX plays a stronger role in the early layers, followed by a reduction in the middle layers, and then a sharp increase in the final layer. In contrast, the V projection shows the opposite trend, i.e., LaX contributes very little in the earlier layers but gradually increases its influence in the later ones. In the attention output projection (Proj), only Layers 1, 10, and 11 exhibit meaningful gate values, indicating that LaX is rarely needed in this component. For the FC1 layer in the MLP, LaX contributions are more evenly distributed across the depth, but still follow a pattern of being higher in the initial and final layers. Finally, in FC2, LaX is largely unused throughout the network, except in Layers 4 and 11, where its contribution becomes significant.

## B.2  Tensor Train Networks with CoLA-style Activations

We explored how LaX interacts with CoLA-style activations in Tensor Train models by injecting a nonlinearity after the down projection. The Tensor Train models and the training script used in this part are identical to Tab. 1. Notably, combining LaX with CoLA leads to a synergistic effect, outperforming both standard TT models and SVD-based low-rank baselines:

| Model | LaX | CoLA Activation | Test Acc (%) |
|---|---|---|---|
| Tensor Train | ✗ | ✗ | 71.11 |
| Tensor Train | ✓ | ✗ | 75.43 (+4.32) |
| Tensor Train | ✗ | ✓ | 72.60 (+1.49) |
| Tensor Train | ✓ | ✓ | **77.73 (+6.62)** |
| SVD | ✓ | ✗ | 77.20 |
| Dense ViT | ✗ | ✗ | 76.74 |

Table 20: How LaX Interacts with CoLA Activation in Tensor Train Networks

## B.3  Ablation Study on Intra-LaX and Inter-LaX

To better understand the source of performance gains, we separately examined the impact of Inter-LaX and Intra-LaX on model accuracy.

We observe that Inter-LaX serves as the primary source of performance gains in Tensor Train models. Moreover, when Intra-LaX is applied selectively, MLP blocks benefit more than Attention blocks.

| Model | Inter-LaX Applied To | Intra-LaX Applied To | Test Acc (%) |
|---|---|---|---|
| Tensor Train | ✗ | ✗ | 71.11 |
| Tensor Train | ✓ | ✓ | 75.43 (+4.32) |
| Tensor Train | ✗ | ✓ | 73.25 (+2.14) |
| Tensor Train | ✓ | ✗ | 73.87 (+2.76) |
| Tensor Train | ✓ | Attention Only | 74.38 (+3.27) |
| Tensor Train | ✓ | MLP Only | **74.98 (+3.87)** |

Table 21: Effect of Inter-LaX and Intra-LaX on Model Performance

| LayerNorm | Training Loss | Test Acc |
|---|---|---|
| ✓ | 2.486 | 77.20 |
| ✗ | nan | 0.10 |

Table 22: Effect of Removing LayerNorm from LaX

## B.4 Ablation Study on Normalization

As expected, removing LayerNorm from LaX causes training to fail immediately. LayerNorm plays a critical role in stabilizing activation distributions and mitigating gradient explosion/vanishing. Without this normalization, activation statistics drift over time, leading to severe gradient instability and ultimately causing the training process to collapse.

## B.5 Ablation Study on Layer Types

We further conducted pre-training experiments to investigate the impact of applying LaX to different layer types. Specifically, we evaluated this on the pre-training of the SVD-ViT-B model on ImageNet-1K.

| Where LaX is Applied | Test Acc (%) |
|---|---|
| None | 75.20 |
| Q only | 75.54 (+0.34) |
| K only | 75.43 (+0.23) |
| V only | 75.91 (+0.71) |
| Attention Block only | 76.38 (+1.18) |
| MLP only | 76.09 (+0.89) |
| Attention Block + MLP | **77.20 (+2.00)** |

Table 23: Ablation study of applying LaX to different layer types.

As shown in the Table, applying LaX to the V projection yields the largest performance gain among the individual attention components. When comparing broader structures, applying LaX to the entire Attention Block provides slightly greater benefits than applying it only to the MLP. Furthermore, the improvements appear to be additive.

