# OpenReview forum: "LaX: Boosting Low-Rank Training of Foundation Models via Latent Crossing"
_NeurIPS.cc/2025/Conference — NeurIPS 2025 poster_

### Official Review · Reviewer_CLBm · 2025-06-29

**Clarity:** 2
**Significance:** 3
**Originality:** 3
**Rating:** 4
**Confidence:** 5

**Summary:**

This paper aims to address the common bottleneck in low-rank compressed deep neural models, where lower-rank models lead to lower performance, but higher-rank models increase the computational burden. To overcome this, the authors propose the LaX method as a plug-in module to enable information flow between low-rank subspaces, inspired by the Residual Mechanism. Moreover, the paper introduces the LaX Gate to align latent spaces with different dimensions. In the experiment, the paper evaluates the effectiveness of LaX through a set of low-rank pre-training and fine-tuning experiments.

**Questions:**

How LaX works on tensor decomposition is unclear, and it lacks a formal description, like its matrix counterpart. Specifically, regarding the Intra-Layer LaX:

• In Figure 2(C), the input is a 2-order tensor, which can first contract with either $C^0$ or $C^1$. What is the contraction order? Assuming the input is a matrix of size $m \times n$ and contracts with $C^0$ first, then after the contraction, will the rank $r_0$ be boosted to $nr_0$?

• In Row 131, the authors mention that a series of latent features will be produced in order: $C^{0}x$, $C^{0}C^{1}x$. According to this description, it seems like the input is just a vector. However, in many cases, the input should also be tensorized, so it should contract with $C^{0}$, $C^{1}$, and $C^{2}$ simultaneously. Therefore, I do not understand how the Intra-Layer LaX is performed, especially when the input is a high-order tensor and contracts with $C^{0}$, $C^{1}$, and $C^{2}$.

• Most of the experimental results are used to support how LaX works on the matrix decomposition-based compression model, with only Table 1 demonstrating how LaX works on the tensor decomposition-based model. The work would benefit from more experiments demonstrating how LaX works on tensor decomposition-based methods.

• As an extension to question 3, tensor decomposition has a more complex structure than matrix decomposition. If the paper could include some additional experiments to better understand how LaX works on tensor decomposition—specifically the pros and cons, and when it is most useful or fails—it would greatly enhance the impact of the paper.

**Ethical Concerns:**

["NO or VERY MINOR ethics concerns only"]

**Final Justification:**

This paper proposes a novel residual-driven mechanism to improve the performance of low-rank compression models. In the rebuttal phase, the authors provided additional explanations that helped clarify the paper’s formulation and addressed most of my concerns. Therefore, I have increased my confidence score. However, I find the experimental validation still somewhat limited.

**Limitations:**

yes

**Paper Formatting Concerns:**

I have reviewed the formatting and have no concerns.

**Quality:**

3

**Strengths And Weaknesses:**

### **Strengths**:

• Low-rank methods for compressing deep neural networks have proven to be an effective way to improve the efficiency of DNNs. However, the performance gap between the full model and low-rank models has always been an important problem to solve. This paper proposes a simple yet effective method that is widely suitable for many different types of low-rank models. The motivation drawn from the Residual Mechanism is interesting, and the implementation is original.

### **Weaknesses**:

• Some parts of how the model works are not described clearly enough, and the paper lacks further experiments to deepen our understanding of the proposed LaX module. Please refer to the questions for more details.

---

> ### Author Rebuttal · Authors · 2025-07-30
>
> Thank you for your meaningful comments! We sincerely appreciate your thoughtful feedback and hope that our responses below will help address your questions:
>
> ### **1. Clarification regarding tensor operations**
>
> ---
>
> Thank you for raising this important point! We will include a more detailed explanation in the revised version. Let us walk through a simple example to clarify here: consider an input $\mathbf{x} \in \mathbb{R}^{B \times d}$, where $B$ is the batch size. We reshape $\mathbf{x}$ into $\mathbb{R}^{B \times d_0 \times d_1}$ to match the first two TT-cores: $\mathbf{C}^0 \in \mathbb{R}^{1 \times d_0 \times r_0}$, $\mathbf{C}^1 \in \mathbb{R}^{r_0 \times d_1 \times r_1}$, within a 4-core Tensor Train: $\{\mathbf{C}^0, \mathbf{C}^1, \mathbf{C}^2, \mathbf{C}^3\}$ setting.
>
> - The TT output $\mathbf{y}_{\text{TT}} = \mathbf{xC^0C^1C^2C^3}$ consists of two stages: (1) **down-projection** via $\mathbf{C}^0, \mathbf{C}^1$ and (2) **up-projection** via $\mathbf{C}^2, \mathbf{C}^3$. Contraction order within each stage is flexible, but for simplicity, we follow the natural order (0→1→2→3),i.e., we first contract $\mathbf{x}$ with $\mathbf{C}^0$, eliminating the $d_0$ mode and producing $\mathbf{xC^0} \in \mathbb{R}^{B \times d_1 \times r_0}$, then we contract $\mathbf{C}^1$ with $\mathbf{xC^0}$, eliminating $d_1$ and $r_0$ and producing $\mathbf{xC^0C^1} \in \mathbb{R}^{B \times r_1}$.
> - Each core $\mathbf{C}^i$ is contracted with intermediate output $\mathbf{xC^0\cdots C^{i-1}}$, so there is no case where multiple cores are contracted with the same $\mathbf{x}$ simultaneously.
> - To illustrate **intra-layer LaX**, we continue from $\mathbf{xC^0C^1} \in \mathbb{R}^{B \times r_1}$ and contract it with $\mathbf{C}^2$, yielding $\mathbf{xC^0C^1C^2} \in \mathbb{R}^{B \times d_2 \times r_2}$. If the shapes align (i.e., $d_1 r_0 = d_2 r_2$), we can fuse $\mathbf{xC^0}\in \mathbb{R}^{B \times d_1 \times r_0}$  with $\mathbf{xC^0C^1C^2}$ directly to build a residual path. If not, we use a gating mechanism to align and merge the features.
>
> ### **2. Additional experiments and discussions on LaX with tensor decomposition**
>
> ---
>
> Thank you for your insightful feedback. We have conducted additional experiments specifically targeting **Tensor Train** models, which we summarize below and plan to include in the revised version.
>
> **2.1 LaX with CoLA Activations in Tensor Train Models**
>
> We explored how LaX interacts with **CoLA-style activations** in TT models by injecting a nonlinearity after the down projection result $\mathbf{xC^0C^1}$. This extension is conceptually natural. Notably, combining **LaX with CoLA** leads to a **synergistic effect**, outperforming both standard TT models and SVD-based low-rank baselines:
>
> | Model | LaX | CoLA Activation | Test Acc (%) |
> | --- | --- | --- | --- |
> | Tensor Train | ✗ | ✗ | 71.11 |
> | Tensor Train | ✓ | ✗ | 75.43 (+4.32) |
> | Tensor Train | ✗ | ✓ | 72.60 (+1.49) |
> | Tensor Train | ✓ | ✓ | **77.73 (+6.62)** |
> | SVD | ✓ | ✗ | 77.20 |
> | Dense ViT | ✗ | ✗ | 76.74 |
>
> This result may suggest that LaX can **amplify the benefits of CoLA-style activations** in tensor format models, significantly boosting the weakest base model to be even stronger than the full-rank model.
>
> **2.2 When Intra-LaX Helps: Pros and Cons**
>
> Understanding where **Intra-LaX** contributes most is crucial, especially for **TT-LaX models with more cores**. To investigate this, we conducted a series of **ablation experiments**, where **✓** indicates that **LaX is applied to all layer types**:
>
> | Inter-LaX Applied To | Intra-LaX Applied To | Test Acc (%) |
> | --- | --- | --- |
> | ✗ | ✗ | 71.11 |
> | ✓ | ✓ | 75.43 (+4.32) |
> | ✗ | ✓ | 73.25 (+2.14) |
> | ✓ | ✗ | 73.87 (+2.76) |
> | ✓ | Attention Only | 74.38 (+3.27) |
> | ✓ | MLP Only | **74.98 (+3.87)** |
>
> These results reveal two key findings:
>
> 1. **Inter-LaX plays a relatively primary role** in performance gains for TT models.
> 2. When applying Intra-LaX selectively, **MLP blocks benefit more than Attention blocks.**
>
> A more granular study, e.g., dissecting Q, K, V projections, is still needed to pinpoint where LaX is most critical. We plan to further investigate this in future work.

---

> > ### Comment · Reviewer_CLBm · 2025-08-03
> >
> > I want to thank the authors for their detailed response. The additional explanations are very helpful and address most of my concerns.

---

> > > ### Author Response · Authors · 2025-08-03
> > >
> > > We are glad to hear that our responses have addressed your concerns. Thank you again for your insightful comments and for taking the time to review our paper!

---

### Official Review · Reviewer_1Hwt · 2025-07-03

**Clarity:** 3
**Significance:** 3
**Originality:** 3
**Rating:** 3
**Confidence:** 3

**Summary:**

This paper introduces Latent Crossing (LaX), a lightweight, plug-and-play module that enhances low-rank training for foundation models like ViTs and LLMs. By injecting residual connections between latent features across or within low-rank layers, LaX improves cross-subspace information flow and effectively expands model capacity without increasing parameter count. It also includes flexible gating mechanisms to manage dimension mismatches. Experiments across pre-training and fine-tuning settings demonstrate consistent performance gains with negligible computational overhead.

**Questions:**

N/A

**Ethical Concerns:**

["NO or VERY MINOR ethics concerns only"]

**Quality:**

3

**Strengths And Weaknesses:**

Strength:
1. Lightweight and parameter-free design with negligible computational overhead.
2. Compatible with various low-rank methods and architectures (e.g., ViT, LLM).
3. Demonstrates consistent performance gains across pre-training and fine-tuning tasks.

Weakness
1. LaX primarily integrates residual fusion into the LoRA setting, with minimal architectural changes and limited theoretical advancement in low-rank modeling.
2. The method is only evaluated under fixed-rank settings; its compatibility with adaptive or dynamic-rank methods (e.g., AdaLoRA) remains unexplored.
3. Performance gains on LLM fine-tuning tasks are relatively modest (see Tables 7–8), and the paper lacks direct comparison with other strong PEFT baselines in this setting.

---

> ### Author Rebuttal · Authors · 2025-07-30
>
> We sincerely thank the reviewer for their thoughtful and constructive feedback. We would like to take this opportunity to clarify that **LaX is not designed solely for Adapters**; rather, it is a **general mechanism for enhancing low-rank models**, applicable to both **low-rank foundation model pre-training from scratch** and **adapter-based fine-tuning** scenarios. We hope that our point-by-point responses below will address any remaining concerns and further clarify the scope and impact of our contributions.
>
> ### **1. Clarification of LaX's broader impact and theoretical justification**
>
> ---
>
> Thank you for the helpful comments. We provide clarifications on the two points raised:
>
> - **We address  a major challenge in low-rank pre-training.** Pre-training low-rank foundation models is fundamentally different from LoRA-style fine-tuning. Rather than inserting adapters, these models **replace dense layers entirely with low-rank layers** and are trained **from scratch**. We would like to emphasize that **enabling effective low-rank model pre-training is a key contribution of LaX**. While such low-rank models typically suffer from performance degradation due to limited capacity, LaX consistently helps **recover**, and **even surpass**, the performance of full-rank models (Table 1, 2, 5, 6), using **2–3× fewer parameters** overall and with negligible computational overhead.
> - **Theoretical analysis**: As stated in the paper, our primary motivation is to improve the capacity of low-rank layers *without explicitly increasing their actual matrix rank*. However, we do offer a theoretical justification, complemented by practical evidence, showing that **LaX increases the *effective rank[1][2][3][4]*** after being applied to the SVD layer. Specifically:
>
> $$
> \mathrm{sr}(\mathbf{W}^{\text{LaX}}) \approx \left( \frac{\sqrt{2d_{\text{out}}} + \sqrt{2d_{\text{in}}}}{\sqrt{d_{\text{out}}} + \sqrt{2d_{\text{in}}}} \right)^2 \mathrm{sr}(\mathbf{W}^{\text{SVD}}) > \mathrm{sr}(\mathbf{W}^{\text{SVD}})
> $$
>
> This expression is derived from concentration results of the Frobenius and spectral norms. In the simulation, we train a simple SVD/LaX layers with identical encoders and decoders on the CIFAR-10 dataset. Due to rebuttal constraints, we are only able to present how the **amplification ratio** $\frac{\mathrm{sr}(\mathbf{W}^{\text{LaX}})}{\mathrm{sr}(\mathbf{W}^{\text{SVD}})}=\left( \frac{\sqrt{2d_{\text{out}}} + \sqrt{2d_{\text{in}}}}{\sqrt{d_{\text{out}}} + \sqrt{2d_{\text{in}}}} \right)^2$**evolves during training** in tabular format. We can see that **SVD w/ LaX always has a higher effective rank than w/o LaX**.
>
> | Epoch | 1 | 20 | 40 | 60 | 80 | 100 |
> | --- | --- | --- | --- | --- | --- | --- |
> | Amplification Ratio | 114.12% | 111.69% | 116.42% | 108.22% | 108.75% | 110.00% |
> | Loss w/o LaX | 2.05 | 1.42 | 1.31 | 1.21 | 1.13 | 1.11 |
> | Loss w/ LaX | 2.05 | 1.39 | 1.23 | 1.10 | 1.00 | 0.97 |
>
>
> ### **2. Comparison with other strong baseline on LLM fine-tuning**
>
> ---
>
> Thank you for pointing this out. We agree that integrating LaX with more advanced methods is a meaningful direction.
>
> - We would like to clarify that **we intentionally chose standard mainsteam methods in both pre-training and fine-tuning** in order to **isolate and clearly demonstrate the effect of LaX** without interference from additional techniques.
> - Regarding other PEFT baselines/adaptive rank methods, LaX is technically compatible with many existing approaches. For example, in AdaLoRA, the LaX Gate can align low-rank spaces with varying shapes. Notably, while existing adaptive low-rank pre-training methods typically fall short of full-rank performance, LaX enables vanilla low-rank models to surpass the full-rank baseline. Due to the limited time window,  we focused on one strong PEFT baseline, **DoRA** [5], and show the comparative results below. **DoRA + LaX consistently performs better or same than plain DoRA in 7 out of 8 tasks.**
>
> | Method | BoolQ | PIQA | SIQA | HellaSwag | WinoGrande | ARC-e | ARC-c | OBQA | Avg. |
> | --- | --- | --- | --- | --- | --- | --- | --- | --- | --- |
> | DoRA | 69.9 | 82.1 | 78.8 | 83.2 | 80.6 | 80.6 | 65.1 | 78.4 | 77.3 |
> | DoRA + LaX | 69.9 (+0.0) | 82.1 (+0.0) | 77.9 (-0.9) | 87.6 (+4.4) | 81.1 (+0.5) | 81.7 (+1.1) | 66.2 (+1.1) | 78.4 (+0.0) | 78.1 (+0.8) |
>
> The experiment is conducted on Llama-7B with an adapter rank=16, using the same hyperparameters as in [5].
>
> **References:**
>
> [1] Vershynin R. High-dimensional probability: An introduction with applications in data science[M]. Cambridge university press, 2018.
>
> [2] Tropp J A. An introduction to matrix concentration inequalities[J]. Foundations and Trends® in Machine Learning, 2015, 8(1-2): 1-230.
>
> [3] Rudelson M, Vershynin R. Sampling from large matrices: An approach through geometric functional analysis[J]. Journal of the ACM (JACM), 2007, 54(4): 21-es.
>
> [4] Ipsen I C F, Saibaba A K. Stable rank and intrinsic dimension of real and complex matrices[J]. arXiv preprint arXiv:2407.21594, 2024.
>
> [5] Liu S Y, Wang C Y, Yin H, et al. Dora: Weight-decomposed low-rank adaptation[C]//Forty-first International Conference on Machine Learning. 2024.

---

> > ### Comment · Area_Chair_dBGk · 2025-08-05
> > **Please respond to the rebuttal and acknowledge the authors**
> >
> > Please respond to the rebuttal and acknowledge the authors

---

> ### Author Response · Authors · 2025-08-07
> **Reminder about the deadline of the extended author-reviewer discussion**
>
> Dear Reviewer **1Hwt**,
>
> Thanks a lot for providing valuable comments to our paper.
>
> We understand that you may have a very busy schedule at this moment. As the extended deadline of the author-reviewer discussion is approaching, we would like to send you a gentle reminder according to the recent NeuRIPS email instruction.
>
> We would highly appreciate it if you can look at our rebuttal and let us know whether your concerns have been addressed or not.
>
>
> Sincerely,
>
> The authors

---

### Official Review · Reviewer_QiUF · 2025-07-03

**Clarity:** 3
**Significance:** 3
**Originality:** 2
**Rating:** 5
**Confidence:** 4

**Summary:**

This paper proposes Latent Crossing (LaX), a simple plug-and-play module that enhances low-rank training methods by enabling information flow between low-rank subspaces through residual connections. The authors show that LaX consistently improves performance across several low-rank methods (SVD, Tensor Train, CoLA) on both vision and language tasks, with minimal parameter overhead.

**Questions:**

- Could you run multiple experiments with different seeds to increase the number of training curves in Figures 6, 7, 8, 9 and to include confidence bounds in Tables 2, 6, 7, 8?
- Have you investigated where in the network LaX provides the most benefit?
- Can you provide any theoretical insight into why LaX improves optimization/generalization in low-rank settings?
- How does LaX interact with other techniques for improving low-rank training? (e.g. Q-LoRA, LoRA-FA, ...)

**Ethical Concerns:**

["NO or VERY MINOR ethics concerns only"]

**Final Justification:**

The authors introduce LaX, a method that introduces residual connections between the LoRA spaces.

Although the presented method is extremely simple (a strength in its easiness to deploy), the authors show that LaX is a consistent improvement over the baselines, especially when taking into account the minimal overhead added.

The authors have addressed the weaknesses we exposed, and we are pleased with the additions to the revision of the manuscript. We recommend accepting the paper.

**Limitations:**

- **Modest improvements in some cases**: While improvements are consistent, they can be modest (e.g. +0.27% for CoLA on 130M LLM). The cost-benefit trade-off isn't always clear.
- **Dependency on baseline quality:** LaX appears to help weaker methods more (e.g. Tensor Train) but provides diminishing returns on already strong methods like CoLA.
- **Gate selection**: the optimal gate type (Identity/Linear/Tensor/Dense) appears to be task and model dependent, requiring empirical search rather than principled selection.

**Quality:**

3

**Strengths And Weaknesses:**

# Strengths
- **Clear motivation and simple solution**: the authors propose a new and intuitive method to improve the training of low-rank models, taking inspiration from ResNet-style residual connections.
- **Comprehensive experimental validation**: the validation spans multiple domains (vision and language), model scales (up to 1B parameters), and different low-rank methods. Results show consistent improvements, with notable gains for Tensor Train methods on ViTs (+4.32%).
- **Practical applicability**: LaX works as a drop-in module for existing low-rank methods, making it easy to adopt in practice.
- **Reproducibility**: the authors will release the code with the paper and have included the hyperparameters of the experiments in the Appendix.


# Weaknesses
- **Novelty**: while LaX proves to be effective, the novelty of the method is rather limited since the core contribution is conceptually simple: adding residual connections in the latent space of the low-rank modules.
- **Limited analysis of when LaX helps most**: while the results show LaX improves the performance the most on weaker baselines (Tensor Train), there's insufficient analysis of what factors determine the magnitude of the improvement.
- **Lack of theoretical analysis**: the paper provides no theoretical justification as to why LaX improves the training of low-rank models.
- **Statistical significance**: the authors just train one model for each setting, and do not include confidence intervals. The small improvement over the baselines could be in part explained by the inherent randomness of the training process.
- **Missing ablations**: the paper could benefit from more ablation studies, such as the effect of applying LaX to different layer types or the impact of normalization in LaX.



## Citations
- The LaX Gates are tightly related to the [*Highway Networks*](https://arxiv.org/abs/1505.00387) , but the authors fail to cite this work.
## Experimental concerns
- The ViT experiments use relatively small models. Do gains persist for bigger models like ViT-H/Vit-G?
- Missing comparisons with other modern LoRA variants, and lack of exploration of orthogonal benefits with these approaches



## Minor issues
- line 113: incorrect grammar "our motivation is **that**, can we improve the performance..."
- line 120: "requiring no additional parameters or computation overhead". Isn't LaX increasing the FLOPs in the forward pass by $(L - 1)h_{in}$, where $L$ is the number of layers? Also, $h_{i-1}$ needs to stay in memory for the computation of the next layer. This is, in fact, mentioned in lines 157, 178.
- In Table 6, the choice of the cross/check for the identity gate and dense gate usage is unintuitive.

---

> ### Author Rebuttal · Authors · 2025-07-30
>
> We are very grateful with the reviewer’s insightful review. Hope our point-wise responses would further clear things out:
>
> ### **1. Confidence bounds and more training curves.**
>
> ---
>
> We agree that including confidence bounds would provide additional insights. Due to the long pre-training time and limited rebuttal period, running multiple trials for all models (which also include related methods such as ReLoRA, GaLore, SLTrain, LORO, etc) to estimate variance is infeasible. However, to provide a valid proof of concept, we have ran 60M LLaMA CoLA+LaX with 5 random seed, and estimate the 95% confidence interval as [33.15, 33.36].
>
> ### **2. Where LaX benefits mostly.**
>
> ---
>
> Thank you for the insightful question. To assess where LaX is most effective, we analyzed the checkpoint from Table 2 (rank=256), focusing on the **scalar values of the Linear Gates**—with higher values indicating greater contribution from LaX.
>
> | LaX Layer | Q | K | V | Proj | FC1 | FC2 |
> | --- | --- | --- | --- | --- | --- | --- |
> | 1 | 3.222 | 3.825 | 0.254 | 0.472 | 2.966 | 0.003 |
> | 2 | 1.370 | 1.653 | 0.406 | 0.098 | 2.005 | 0.004 |
> | 3 | 1.596 | 1.960 | 0.816 | 0.037 | 1.462 | 0.040 |
> | 4 | 1.550 | 1.522 | 1.412 | 0.033 | 1.360 | 0.552 |
> | 5 | 2.071 | 2.161 | 0.847 | 0.018 | 1.272 | 0.026 |
> | 6 | 2.184 | 2.023 | 1.746 | 0.004 | 1.929 | 0.002 |
> | 7 | 0.533 | 1.210 | 2.116 | 0.002 | 2.038 | 0.004 |
> | 8 | 0.370 | 1.336 | 1.895 | 0.001 | 1.860 | 0.005 |
> | 9 | 0.259 | 1.040 | 1.911 | 0.002 | 1.800 | 0.004 |
> | 10 | 0.857 | 0.955 | 1.356 | 0.869 | 2.332 | 0.002 |
> | 11 | 1.203 | 2.330 | 2.983 | 1.024 | 2.012 | 1.477 |
>
>  Our analysis reveals several interesting observations:
>
> - For the **Q and K projections**, LaX contributes more in the first half of the layers than the latter half, with a sharp increase in gate values observed in the final layer.
> - The **V projection** exhibits a distinct pattern, where LaX contributes less in the earlier layers compared to the later ones.
> - In the **attention output projection layer**, only Layers 1, 10, and 11 show relatively higher gate values, while the remaining layers contribute minimally.
> - For **FC1 in the MLP**, the contribution of LaX is more evenly distributed across layers, but still follows the general trend of higher values in the early and final layers.
> - For **FC2 in the MLP**, only Layers 4 and 11 show meaningful gate values, with others being close to zero.
>
> We thank the reviewer once again for their insightful observations, which may point toward a promising future direction—**a sparse variant of LaX** that selectively activates only the most impactful layers.
>
> ### **3. How does LaX interact with other techniques for improving low-rank training?**
>
> We fully agree that exploring the interaction between LaX and other techniques for low-rank training is a highly promising direction. In line with your suggestion regarding Q-LoRA, we’d like to share some preliminary results from an ongoing work investigating the integration of **LaX with quantization methods**.
>
> - We pre-train ViT-B and its CoLA + LaX variant (w/ Tensor Gate) at FP32 on Imagenet-1k, then we perform a series of simple post-training quantization (Optimum Quanto from Huggingface for linear quantization) on the resulting model,
>
> | Precision | Model |  # Params (M) | Test Acc |
> | --- | --- | --- | --- |
> | FP32  | ViT-B | 86.56  | 76.74 |
> | FP32  | CoLA | 44.17 | 76.04 |
> | FP32  | CoLA + LaX | 44.24   | 77.84 |
> | Int8  | CoLA + LaX | 44.24   | 77.70 |
> | Int4  | CoLA + LaX | 44.24   | 76.57 |
> - As shown, **LaX integrates effectively with quantization techniques**. Notably, Int4 CoLA + LaX achieves performance comparable to the full-precision dense ViT-B model, while achieving a **16× reduction in model size** (**2× in parameters** and **8× in precision**).
>
> ### **4. Ablation for applying LaX to different layer types and for Normalization.**
>
> ---
>
> We conducted the suggested ablation experiments specifically on the **pre-training of the SVD-ViT-B model** on **ImageNet-1K**.
>
> - Ablation for applying LaX to different layer types:
>
> | Where LaX | Test Acc  |
> | --- | --- |
> | - | 75.20 |
> | Q | 75.54 (+0.34) |
> | K | 75.43 (+0.23) |
> | V | 75.91 (+0.71) |
> | Attn. Block | 76.38 (+1.18) |
> | MLP | 76.09 (+0.89) |
> | Attn. Block + MLP | 77.20 (+2.00) |
>
> As shown, applying LaX to the V-proj yields the most notable gains among attention layers. Between MLP and Attention Blocks, the latter benefits slightly more. Notably, the benefit of LaX appears additive, with multiple applications leading to cumulative improvements.
>
> - Ablation for Normalization:
>
> Removing LayerNorm (LN) from LaX causes training to **fail from the start, as expected**. LN stabilizes activation distributions and prevents gradient instability; without it, training collapses due to shifting statistics.
>
> | LN | Training Loss | Test Acc |
> | --- | --- | --- |
> | w/ | 2.486 | 77.20 |
> | w/o | nan | 0.1 |
>
> ### **5. Dependency on baseline quality. Modest improvements in some cases. The cost-benefit trade-off isn't always clear.**
>
> ---
>
> Thank you for your valuable comments. As discussed in the paper:
>
> - **LaX is specifically designed to recover the performance lost in low-rank models relative to their full-rank counterparts.** Naturally, when starting from a strong low-rank baseline, such as CoLA, the improvement is less significant. Nonetheless, the integration of LaX consistently enables CoLA to surpass the full-rank baseline in both vision and language modeling tasks.
> - **LaX introduces minimal computational overhead**, even with the most expensive practical variant. As noted in the practical guideline section, **Tensor Gate** is the most costly one in practice. The **relative FLOPs overhead**, compared to the baseline low-rank model,
>
> | Gating Type | ViT-B | ViT-L |
> | --- | --- | --- |
> | **Tensor Gate** | 1/48 | 1/64 |
>
> ### **6. The optimal gate type appears to be task and model dependent, requiring empirical search rather than principled selection.**
>
> ---
>
> We appreciate the reviewer’s concern and agree that different gate types yield different trade-offs. However, we would like to clarify that the core motivation of the gate design in LaX is to **align latent dimensions across or within low-rank layers,** which is not mandatory in many cases, especially with SVD and CoLA. Without gating, **LaX still yields consistent gains across various domains,** as shown in Table 2 and Table 6.
>
> ### **7. Novelty/theoretical analysis.**
>
> ---
>
> Thank you for the thoughtful comment. While LaX incorporates residual connections, its **core novelty lies in addressing a unique challenge in low-rank adaptation**. As stated in the paper, our primary motivation is to **enhance the capacity of low-rank layers without increasing their matrix rank**.
>
> - Importantly, the **effective rank $\mathrm{sr}=\frac{\mathbf{||W||}^2_F}{||\mathbf{W}||^2_2}$** of a layer is often lower than its matrixl rank. This observation opens up an opportunity: improve the effective rank in a lightweight and principled way — latent feature fusion.
> - We support this intuition with both **theoretical justification** under assumptions and **empirical evidence** during the actual training process, demonstrating that LaX increases the effective rank and leads to consistent performance gains. While the mechanism may resemble residual connections in form, LaX is functionally and conceptually distinct in its purpose. Specifically (notion is the same as in our paper),
>
> $$
> \mathrm{sr}(\mathbf{W}^{\text{LaX}}) \approx \left( \frac{\sqrt{2d_{\text{out}}} + \sqrt{2d_{\text{in}}}}{\sqrt{d_{\text{out}}} + \sqrt{2d_{\text{in}}}} \right)^2 \mathrm{sr}(\mathbf{W}^{\text{SVD}}) > \mathrm{sr}(\mathbf{W}^{\text{SVD}})
> $$
>
> - In the simulation, we train a simple SVD/LaX layers with identical encoders and decoders on the CIFAR-10. Due to rebuttal constraints, we are only able to present how the **amplification ratio** $\frac{\mathrm{sr}(\mathbf{W}^{\text{LaX}})}{\mathrm{sr}(\mathbf{W}^{\text{SVD}})}$**and training loss** **evolve during training** in tabular format. We can see that **SVD w/ LaX always has a higher effective rank than w/o LaX**.
>
> | Epoch | 1 | 20 | 40 | 60 | 80 | 100 |
> | --- | --- | --- | --- | --- | --- | --- |
> | Amplification Ratio  | 114.12% | 111.69% | 116.42% | 108.22% | 108.75% | 110.00% |
> | Loss w/o LaX | 2.05 | 1.42 | 1.31 | 1.21 | 1.13 | 1.11 |
> | Loss w/ LaX | 2.05 | 1.39 | 1.23 | 1.10 | 1.00 | 0.97 |
>
> ### **8. Bigger models like ViT-H/G.**
>
> ---
>
> We agree that studying the **scaling behavior of LaX combined with low-rank models** is an important direction. Due to budget constraints, we were unable to pre-train many large models. But we have included **four pre-training results on 1B-scale LLMs** in Table 5, which serve as meaningful references for a scale positioned between ViT-H (\~650M) and ViT-G (1.8B). At 1B scale, LaX still consistently brings benefits to all the low-rank models.
>
> ### **9. More LoRA-like variants.**
>
> ---
>
> For low-rank fine-tuning, we adopt LoRA as the base method because it is the foundational approach in this domain and we aim to isolate the impact of LaX itself, just as in our pre-training experiments. However, in order to explore LaX’s orthogonal benefits when combining with other modern LoRA variants, we added addtional experiments on **DoRA** [1], and show results below. Notably, **DoRA + LaX consistently performs better or same than plain DoRA in 7 out of 8 tasks.**
>
> | Method | BoolQ | PIQA | SIQA | HellaSwag | WinoGrande | ARC-e | ARC-c | OBQA | Avg. |
> | --- | --- | --- | --- | --- | --- | --- | --- | --- | --- |
> | DoRA | 69.9 | 82.1 | 78.8 | 83.2 | 80.6 | 80.6 | 65.1 | 78.4 | 77.3 |
> | DoRA + LaX | 69.9 (+0.0) | 82.1 (+0.0) | 77.9 (-0.9) | 87.6 (+4.4) | 81.1 (+0.5) | 81.7 (+1.1) | 66.2 (+1.1) | 78.4 (+0.0) | 78.1 (+0.8) |
>
> The experiment is conducted on Llama-7B with an adapter rank=16, using the same hyperparameters as in [1].
>
> **References:**
> [1] Dora: Weight-decomposed low-rank adaptation.

---

> > ### Comment · Reviewer_QiUF · 2025-08-06
> >
> > We deeply thank the authors for their thorough response, and apologize for our late reply.
> >
> > **Confidence bounds and more training curves**: we acknowledge that providing confidence bounds on all experiments is infeasible within the rebuttal period, and are happy to see that in the small-scale experiment the confidence interval is 95%. We believe the confidence is high enough and that the ablation with 5 different seeds is sufficient.
> >
> > **Where LaX benefits mostly**: we thank the authors for showing the contribution of LaX on different layers and submodules. We believe that the insights from this experiment add positively to their research, and gives a better understanding of where LaX works best.
> >
> > **How does LaX interact with other techniques for improving low-rank training?**: we thank the authors for exploring the combination of LaX with CoLA. By showing that LaX performance holds with quantization techniques, it shows LaX as a promising method for real world deployment.
> >
> > **Ablation for applying LaX to different layer types and for Normalization**: acknowledged.
> >
> > We sincerely thank the reviewers for their clarifications.
> >
> > We are convinced that these additions to the manuscript will strengthen the author's contributions, and we will update our scores accordingly.

---

> > > ### Author Response · Authors · 2025-08-06
> > >
> > > We sincerely thank the reviewer for their insightful comments and acknowledgment. We will incorporate these points into the revised version of our paper. Thank you again for taking the time to review our work!

---

### Official Review · Reviewer_bjrv · 2025-07-09

**Clarity:** 4
**Significance:** 4
**Originality:** 3
**Rating:** 5
**Confidence:** 4

**Summary:**

The authors introduce Latent Crossing (LaX), a small addition to many low-rank modules/PEFT methods that improves performance across the board. LaX involves equipping the latent space of low-rank modules with residual connections, followed by layer normalization. Before applying the residual connection, however, one of four proposed gating layers is applied to the low-rank representation. The proposed layers are lightweight in terms of computation and memory complexity, especially for the PEFT setting, where these considerations are less important. To validate the effectiveness of their method, the authors provide extensive experiments across ViT pre-training, LM pre-training, and LLM fine-tuning. The results generally show that the author’s proposed method improves performance beyond.

**Questions:**

- In Table 2, the Identity gate has the same number of parameters as the Base model, but the authors state on line 164 that they postpend Layer Normalization to each residual connection. Don’t the layer norms have parameters?

- In Table 4, why does CoLa + LaX outperform Full-rank?

- Do you think the result from table 4 would hold if a different optimizer, e.g., Muon, were used, which might be able to take advantage of the full rank matrix?

**Ethical Concerns:**

["NO or VERY MINOR ethics concerns only"]

**Final Justification:**

I have maintained my score. I recommend accepting the paper.

**Issues resolved**
- The reason for CoLa + LaX outperforming Full-rank pre-training.
- More information regarding hyperparameter selection choices.
- Discussion of an additional related work.

**Limitations:**

Yes

**Quality:**

3

**Strengths And Weaknesses:**

**Strengths**:
- The paper is well written and easy to follow.
- The tables are clear and provide all the necessary details
- The method outperforms all baselines tested within the author’s study.
- The authors provide training curves for their method in the appendix, which is appreciated.

**Weaknesses**:
- The authors show a surprising result: pre-training with CoLa + LaX outperforms Full-rank pre-training (see Table 4), but this is not explained in the text, leading me to wonder if the hyperparameters for Full-rank were well tuned?
- In section A.3 of the appendix, the authors state that they sweep at the scale of 1e-3 for base models and 1e-2 for LaX variants. For reproducibility, it would be best to mention how many HP configurations were tried for each method and ideally the values shown.
- When browsing related literature, I found a closely related work published last year that should be discussed [1]. It does not overlap with the author's method, but could complement it and is close enough to warrant discussion.



When browsing related literature, I also found [2], a closely related concurrent work which might be of use to the authors, but that is too new to require evaluation.

---
**References**

[1][ResLoRA: Identity Residual Mapping in Low-Rank Adaption; ACL 2024]

[2][LoR2C : Low-Rank Residual Connection Adaptation for Parameter-Efficient Fine-Tuning.]

---

> ### Author Rebuttal · Authors · 2025-07-30
>
> We really appreciate the reviewers’ thoughtful review and recognition of our contributions. Hope our point-wise responses would further clear things out:
>
> ### **1. Clarification of why CoLA+LaX outperforming full-rank baselines**
>
> ---
>
> Thanks for your comments. We confirm that **CoLA + LaX outperforms the dense model in both Vision** (Table 1, we train both full-rank and low-rank models using configurations from [1]) and **Language model pre-training** (Table 5, we directly compare LaX with baselines reported in related work [Galore, SLTrain, CoLA, LORO]). According to the CoLA paper, CoLA improves architectural efficiency by replacing an arbitrary linear operator by a composition of linear-nonlinear-linear operator, which is theoretically capable of approximating any continuous function, powered by the universal approximation theorem. We speculate that such a deeper, nested structure might empower CoLA with a higher capacity while using fewer parameters, and LaX further boosts its performance, therefore surpassing full-rank models.
>
> ### **2. Clarification of the parameters introduced by LaX’s layer norm**
>
> ---
>
> Thanks!  We would like to clarify that the LayerNorm modules introduced in LaX indeed introduce parameters to the base model. However, **the increase is negligible** — only $2 r$ parameters per layer, $\frac{1}{d}$ of the base model. Given that $d$ is normally very large, the overall increase is trivial and does not visibly change the reported parameter counts due to rounding. We appreciate the opportunity to clarify this point and will consider explicitly noting it to avoid confusion.
>
> ### **3. Do authors think CoLA+LaX can still outperform full-rank models if using other optimizers, such as Muon?**
>
> ---
>
> Thank you for your insightful comment! It is possible that a fine-tuned optimizer might further improve the accuracy of either CoLA+LaX, full-rank models or the both. In this paper, we demonstrate the performance boost of LaX while using mainsteam optimizer such as Adam. Definitely many other research opportunities can be inspired by considering the combinations of various optimizers and architectures.
>
> ### **4. Discussion regarding ResLoRA/LoR2C.**
>
> ---
>
> We thank the reviewer for highlighting these relevant and recent works. Both are targeting the **gradient vanishing/explosion challenges in PEFT**. We acknowledge their potential relevance and see **promising opportunities for incorporating their insights into future extensions** of our approach.
>
> - **ResLoRA** tackles gradient propagation issues in LoRA by introducing **shortcut connections at various positions within the adapter structure**, supported by a **mathematical analysis from an optimization perspective**. **LoR2C** mitigates gradient vanishing by introducing **residual connections across entire blocks with an extra LoRA block.**
> - In LaX, we currently apply LaX between the **same type of layers** across transformer blocks, guided by a prior on semantic alignment. However, their work suggests that exploring **LaX variants that span different connection types** is possible, particularly in the context of **Tensor Train models.** Owing to their richer and more structured latent spaces, Tensor Train models offer greater flexibility for experimenting with alternative LaX connection patterns. We are looking forward to exploring this in our future research.
> - Moreover, we are also deeply interested in the **theoretical understanding of LaX**. We believe that their optimization-based analyses could inspire new theoretical analyses to explain LaX.
>
> We thank the reviewer once again for pointing out these valuable works, which may serve as complementary directions to LaX in our future research efforts!
>
> **References:**
>
> [1] Dosovitskiy A, Beyer L, Kolesnikov A, et al. An image is worth 16x16 words: Transformers for image recognition at scale[J]. arXiv preprint arXiv:2010.11929, 2020.

---

> ### Comment · Reviewer_bjrv · 2025-08-04
>
> Thank you for the reply, you have addressed almost all my concerns.
>
> As stated in my review, Its quite important for reproducibility to report the values of hyperparameter configurations swept for each method.
>
> Could the authors provide this information?

---

> ### Author Response · Authors · 2025-08-04
> **Follow-up discussion**
>
> Thank you for your response! And we are glad to hear that we have resolved almost all your concerns. We further clarfiy our hyper-parameter configurations as below:
>
> - **ViT Pre-Training Experiments**: As we mentioned in our rebuttal response, we adopted all the hyper-parameters from [1] except the learning rate. The original LR=3e-3 caused unstable training in LaX, therefore we lowered it to 1e-3 and kept this setting for all the experiments.
> - **LLM Pre-Training Experiments**: As we mentioned in our rebuttal response, all baselines results are directly reported from related works (GaLore, SLTrain, CoLA, LORO). And we adopted most hyper-parameters from these baseline methods, except learning rate and gradient clipping scale.
>     - **For gradient clipping**, we simply found using a slightly smaller scale, i.e., 0.5 instead of 1.0, yields relatively better training stabilities, therefore we used this setting for all LLM pre-training experiments.
>     - **For learning rate**, we only swept for small models, such as 60M and 130M. And our early explorations at 60M scale suggested LaX could benefit from significantly larger learning rate, typically >1e-2. Then we tried gradually increasing it until the model diverges, which gave us 4e-2 for 60M scale. Then larger scale models will start from this LR, if it diverges, **we typically just halve the value, and it would yield a very good setup**. If not, then we further halve the value. This is how we found the LR for 130M, 350M and 1B. Sweeping would be too expensive for models >=350M.
> - **LLM Fine-Tuning Experiments**: We simply followed the same setup in [2], without any hyper-parameter tuning or sweeping.
>
> **References:**
>
> [1] Dosovitskiy A, Beyer L, Kolesnikov A, et al. An image is worth 16x16 words: Transformers for image recognition at scale[J]. arXiv preprint arXiv:2010.11929, 2020.
>
> [2] Hu, Zhiqiang, et al. "Llm-adapters: An adapter family for parameter-efficient fine-tuning of large language models." arXiv preprint arXiv:2304.01933 (2023).

---

> > ### Comment · Reviewer_bjrv · 2025-08-04
> >
> > Thank you for the reply. You have addressed all my concerns.
> >
> > If it has not already been done, I would suggest including the above explanation about hyperparameter selection in the paper to improve clarity.

---

> > > ### Author Response · Authors · 2025-08-04
> > >
> > > We are very grateful for your insightful comments and suggestions! We will certainly include these clarifications into our final version. Thank you again for reviewing our paper!

---

### Note · Authors · 2025-08-13

**Dear AC/SAC and reviewers,**

We greatly appreciate your efforts in reviewing this work. Below is our rebuttal summary:

## **Major Problems Addressed**

1. **Why CoLA+LaX outperforms full-rank:** CoLA boosts LLM capacity via a linear–nonlinear–linear structure with fewer parameters; LaX further improves performance by fusing latent features to raise *effective* rank without increasing matrix rank. This synergy lets LaX+CoLA outperform full-rank models in pre-training. New experiments also show strong benefits for Tensor Train layers.
2. **Parameter/compute overhead:** LaX’s LayerNorm adds only $2r$ params per layer ($\frac{1}{d}$ of base model). FLOPs overhead is minimal; even Tensor Gate adds just 1/48 FLOPs for ViT-B and 1/64 for ViT-L over low-rank baselines.
3. **Interaction with other techniques:** Our new results shows LaX’s gains are additive with quantization, DoRA, and Tensor-Train-CoLA.
4. **Fine-grained Ablations:** (1) In ViT, V-projection benefits most; combining attention+MLP is additive. (2) Removing LayerNorm breaks training. (3) For Tensor Train, **Inter-LaX** drives main gains; MLPs benefit more from **Intra-LaX** than attention.
5. **Theoretical Justification:** LaX increases *effective rank* $\frac{||W||^2_F}{||W||^2_2}$ of SVD layers without changing their physical matrix rank, supported by amplification ratio and lower training loss in controlled CIFAR-10 experiments.
6. **Scaling:** Up to **1B-scale LLM pre-training**, LaX continues to boost low-rank models, suggesting promise for ViT-H/G scales.

## **Reviewer Responses**

Reviewers **bjrv**, **QiUF**, and **CLBm** responded with satisfaction. Reviewer **1Hwt** did not join the discussion, but we believe their concerns have been clarified:

- “**LaX mainly integrates residual fusion into LoRA**”: Unlike LoRA, which targets on fine-tuning where low-rank methods typically perform well, LaX is mainly developed for **pre-training, where low-rank methods often suffer from a huge performance drop**. LaX narrows this gap, significantly boosting low-rank pre-training accuracy. The reviewer's claim narrows LaX’s scope and overlooks its broader impact.
- “**LLM fine-tuning gains are modest**”: When applied in fine-tuning, LoRA+LaX matches SOTA PEFT methods like DoRA, and can further boost DoRA to perform even better.

We thank the AC/SAC for encouraging timely, high-quality reviews, making this year’s rebuttal process a pleasant and rewarding experience.

**Best regards,**

The authors

---

### Decision · Program_Chairs · 2025-09-17

**Decision:**

Accept (poster)

**Comment:**

This submission proposes a new method to improve the training of low-rank models, taking inspiration from ResNet-style residual connections. Reviewers found the method intuitive and the experiments demonstrate the method's efficacy. The weaknesses of the paper are that the method is not very novel (i.e., residual connections are a common idea), it lacks theoretical motivation, and the method is only compared against LoRA as a baseline.

Ultimately, I think a simple idea that works is worth publishing, and the only negative reviewer was not engaged at all and did not identify any weaknesses the other reviewers missed. I would strongly encourage the authors to include other baseline PEFT methods that have been shown to improve on LoRA. I would also encourage running the method on multiple random seeds and reporting the variance.